

A manuscript submitted to discussion forum of BG as a BG discussion paper (bg-2016-222):
Manganese and iron reduction dominate organic carbon oxidation in deep
continental margin sediments of the Ulleung Basin, East Sea
Jung-Ho Hyun[1]*, Sung-Han Kim[1], Jin-Sook Mok[1], Hyeyoun Cho[1], Tongsup Lee[2], Verona
Vandieken[3] and Bo Thamdrup[4]*
[1]Department of Marine Sciences and Convergent Technology, Hanyang University, 55
Hanyangdaehak-ro, Ansan, Gyeonggi-do 15588, South Korea
[2]Department of Oceanography, Pusan National University, Busan 609-735, South Korea
[3]Institute for Chemistry and Biology of the Marine Environment, University of Oldenburg,
Carl-von-Ossietzky-Str. 9-11, 26129 Oldenburg, Germany
[4]Nordic Center for Earth Evolution, Department of Biology, University of Southern Denmark,
Campusvej 55, 5230 Odense M, Denmark
*Correspondence to:
Jung-Ho Hyun (hyunjh@hanyang.ac.kr)
Bo Thamdrup (bot@biology.sdu.dk)



**Abstract.** Rates and pathways of benthic organic carbon ($C_{org}$) oxidation were investigated in
surface sediments of the Ulleung Basin (UB) characterized by high organic carbon contents
(> 2.5 %, dry wt.) and very high concentrations of Mn oxides (> 200 µmol $cm^{-3}$) and Fe
oxides (up to 100 µmol $cm^{-3}$). The combination of geochemical analyses and independently
executed metabolic rate measurements revealed that Mn and Fe reduction were the dominant
$C_{org}$ oxidation pathways in the center of the UB, comprising 45 % and 20 % of total $C_{org}$
oxidation, respectively. By contrast, sulfate reduction was the dominant $C_{org}$ oxidation
pathway accounting for 50 % of total $C_{org}$ mineralization in the continental slope. The relative
significance of each $C_{org}$ oxidation pathway matched the depth distribution of the respective
electron acceptors. The relative importance of Mn reduction for $C_{org}$ oxidation displays
saturation kinetics with respect to Mn oxide content with a low half-saturation value of 8.6
µmol $cm^{-3}$, which further implies that Mn reduction can be a dominant $C_{org}$ oxidation process
even in sediments with lower $MnO_2$ content as known from several other locations. This is
the first report of a high contribution of manganese reduction to $C_{org}$ oxidation in offshore
sediments on the Asian margin. The high manganese oxide content in the surface sediment in
the central UB was maintained by an extreme degree of recycling, with each Mn atom on
average being reoxidized ~3800 times before permanent burial. This is the highest degree of
recycling so far reported for Mn-rich sediments, and it appears linked to the high benthic
mineralization rates resulting from the high organic carbon content that indicate the UB as a
biogoechemical hotspot for turnover of organic matter and nutrient regeneration. Thus, it is
important to monitor any changes in the rates and partitioning of $C_{org}$ oxidation to better
understand the biogeochemical cycling of carbon, nutrients and metals associated with long-
term climatic changes in the UB, where the fastest increase in sea water temperature has been
reported for the past two decades.


**Keywords.**  Benthic mineralization, Manganese reduction, Iron reduction, Sulfate reduction,
Ulleung Basin, East Sea



## 1 Introduction

Although they cover only 15 % ($47 \times 10^6$ km$^2$) of the ocean surface area, sediments of continental margins (200 – 2000 m depth) are characterized by enhanced organic matter flux generated either by vertical transport from the highly productive overlying water column or by lateral transport from adjacent shelves, and thus play an important role in deposition and mineralization of organic matter (Romankevich, 1984, Jahnke et al., 1990; Walsh, 1991; Jahnke and Jahnke, 2000). Organic particles that reach the seafloor are quickly mineralized by hydrolysis, fermentation, and a variety of respiratory processes using different electron acceptors such as oxygen, nitrate, Mn oxides, Fe oxides, and sulfate (Froelich et al., 1979; Jørgensen, 2006). The partitioning of organic carbon ($C_{org}$) oxidation among the different electron accepting pathways has profound influence on the distribution and the release and/or retention of Mn, Fe, S and nutrients (nitrogen and phosphate) (Canfield et al., 2005; Jørgensen, 2006). Therefore, it is particularly important to elucidate the contribution of each $C_{org}$ oxidation pathway in order to better understand the role of sediments in biogeochemical element cycles.

The relative significance of each carbon oxidation pathway is largely controlled by the combination of organic matter supply and availability of electron acceptors. In general, aerobic metabolism dominates the organic matter mineralization in deep-sea sediments that are characterized by low organic matter content (Jahnke et al., 1982; Glud, 2008). In contrast, owing to high sulfate concentrations in marine sediment, sulfate reduction accounts for up to 50 % of total carbon oxidation in continental margins with high organic matter flux (Jørgensen, 1982; Jørgensen and Kasten, 2006). However, in sediments where manganese and iron oxides are abundant or rapidly recycled, microbial reduction of manganese and iron can be the dominant electron accepting processes over sulfate reduction (Sørensen and Jørgensen, 1987; Aller, 1990; Canfield et al., 1993b). The significance of dissimilatory iron reduction for $C_{org}$ oxidation is well established in the sediments of various continental margins and coastal wetlands (Thamdrup, 2000; Thamdrup and Canfield, 1996; Jensen et al. 2003, Kostka et al., 2002a, 2002b; Vandieken et al., 2006; Hyun et al., 2007, 2009b). However, only a few locations such as Panama Basin (Aller, 1990), the coastal Norwegian trough in Skagerrak and an adjacent fjord (Canfield et al., 1993a, 1993b; Vandieken et al., 2014), the Black Sea shelf (Thamdrup et al., 2000) and the continental shelf of the northern Barents Sea (Vandieken et al., 2006; Nickel et al., 2008) are known where microbial

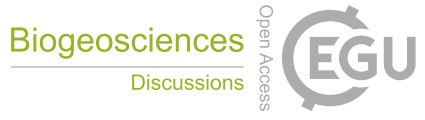

manganese reduction significantly contributes to carbon mineralization.
The East Sea (often referred to as Japan Sea), located in the far eastern part of the Eurasian
continental margin, consists of three major basins deeper than 2,000 m, Japan Basin, Yamato
Basin and Ulleung Basin (Fig. 1). Compared to the other two basins, the Ulleung Basin (UB)
is characterized by higher phytoplankton biomass and primary production (Yamada et al.,
2005; Yoo and Park, 2009), which is associated with coastal upwelling (Hyun et al., 2009a).
The enhanced biological production in the water column is responsible for the high organic
carbon content ( > 2.5 % wt) in the sediment, and the highest rates of $C_{org}$ oxidation
compared to any other deep-sea sediments with similar depth range (Lee et al., 2008; Hyun et
al., 2010). An intriguing geochemical property of the UB surface sediment is the high content
of Mn oxides ( > 200 μmol cm$^{-3}$) and Fe oxides (up to 100 μmol cm$^{-3}$) (Cha et al., 2007;
Hyun et al., 2010). In accordance with these geochemical findings, the suppression of sulfate
reduction (Hyun et al., 2010) and accumulation of $Mn^{2+}$ in anoxic incubation of surface
sediment (Vandieken et al., 2012) strongly implied that the $C_{org}$ oxidation in the surface
sediment of the UB is dominated by microbial manganese and iron reduction, but actual rates
and partitioning of each electron accepting pathway in $C_{org}$ oxidation remain to be determined
in this deep marginal sediment underlying highly productive water column.
The primary objective of this paper was to characterize the sediment biogeochemistry with
regard to the rate of $C_{org}$ oxidation and partitioning of major terminal electron accepting
pathways at two contrasting sites in the continental slope and rise in the UB. Here, for the
first time in sediments of the Asian marginal seas, we document that Mn and Fe reduction are
the dominant $C_{org}$ oxidation pathways accounting for respectively 45 % and 20 % of total $C_{org}$
oxidation in the center of the UB, and suggest that Mn and Fe reduction may be of greater
importance in deep-sea sediments than previously recognized.


**2 Materials and methods**

**2.1 Study site**

Shipboard experiments were conducted in June, 2009 at two sites on the continental slope
(M1) and rise (D3) in the center of the UB (Fig. 1, Table 1). Surface sediments consist of
fine-grained clay with a mean grain size less than 0.004 mm in diameter (Cha et al., 2007).



The two stations were characterized by the two contrasting sediment colors. The Mn oxide-
enriched surface sediment at the basin site (D3) was reddish-brown, whereas at the slope site
(M1) it exhibited the typical gray-brown color of muddy continental sediments (Fig. 1).
Further environmental properties are listed in Table 1.
**2.2 Sampling and handling**
Sediment samples were collected with a box corer. Onboard, duplicate or mostly triplicate
sub-samples for geochemical analyses were collected using acrylic cores (6–9 cm i.d.). The
sub-cores for geochemical analyses were immediately sealed with butyl rubber stoppers and
transferred to a $N_2$-filled glove bag for sectioning and loading into polypropylene centrifuge
tubes that were then tightly capped and centrifuged for 15 min at $5000 \times g$. After
reintroduction into the $N_2$-filled glove bag, pore-waters were sampled and filtered through
0.2-μm cellulose ester syringe filters (ADVANTEC, Toyo Rashi Kaisha, Ltd). One to two
mL of pore water to determine $NH_4^+$ was fixed with saturated $HgCl_2$, and frozen. For
determination of $Fe^{2+}$, Mn, $SO_4^{2-}$ and $Ca^{2+}$, 2 mL of the pore water were acidified with 12M
HCl and stored at 4 ℃. Pore-water for sulfide analysis was preserved with Zn acetate (20 %).
Sediments for solid-phase analysis were frozen at − 25 ℃ for future analyses.
**2.3 Anoxic bag incubations**
Anaerobic carbon mineralization rates and dissimilatory Mn and Fe reduction rates were
determined in batch incubations based on the procedures of Canfield et al. (1993b) and
Thamdrup and Canfield (1996). Sediment cores were transferred to a $N_2$-filled glove bag and
sliced in 2-cm intervals to a depth of 10 cm. Sediment from parallel sections was pooled,
mixed and loaded into gas-tight plastic bags (Hansen et al., 2000). The bags were sealed
without gas space, and incubated in the dark at near in situ temperature (ca. 1 – 2 ℃) in
larger $N_2$ filled bags to ensure anoxic conditions. Over a period of 18 days of incubation, sub-
samples to determine the accumulation of total dissolved inorganic carbon (DIC) and Mn in
pore water were withdrawn on days 0, 1, 3, 5, 9 and 18. Two 50-mL centrifuge tubes per bag
were filled completely with sediment in $N_2$-filled glove bag, and pore-water was extracted as
described above. For DIC analysis, we collected 1.8 mL aliquots into glass vials without head
space, fixed with 18 μL of $HgCl_2$ (125 mM), and stored at 4 ℃ until analysis in 4 weeks.



Samples for Mn analysis were acidified with 12M HCl and stored at 4 ℃. Sediment
remaining after the collection of pore water was frozen at −25 ℃ for later analysis of oxalate
extractable solid Fe(II).

**2.4   Pore-water analyses**

Total dissolved inorganic carbon (DIC) and $NH_4^+$ were measured by flow injection analysis
with conductivity detection (Hall and Aller, 1992). Nitrate was measured
spectrophotometrically (Parsons et al., 1984). Dissolved $Fe^{2+}$ was determined by colorimetric
method with Ferrozine (Stookey, 1970). Dissolved Mn and $Ca^{2+}$ were analyzed in acidified
pore water by inductive coupled plasma-atomic emission spectrometry (ICP-AES, Optima
3300DV, Perkin-Elmer Co.) and flame atomic absorption spectrometer (SpectrAA 220/FS,
Varian), respectively (Thamdrup and Canfield, 1996). Dissolved sulfide was determined by
the methylene blue method (Cline, 1969). Sulfate concentrations were measured using ion
chromatography (Metrohm 761).

**2.5   Solid-phase analyses**

Total oxalate-extractable Fe [Fe(II) + Fe(III)] was extracted from air-dried sediment in a 0.2
M oxic oxalate solution (pH 3) for 4 h (Thamdrup and Canfield, 1996), and Fe(II) was
extracted from frozen sediment in anoxic oxalate (Phillips and Lovley, 1987). The total
oxalate-extractable Fe and Fe(II), hereafter total $Fe_{(oxal)}$ and $Fe(II)_{(oxal)}$, were determined as
described in pore-water analysis of $Fe^{2+}$. Oxalate-extractable Fe(III), hereafter $Fe(III)_{(oxal)}$,
was defined as the difference between total $Fe_{(oxal)}$ and $Fe(II)_{(oxal)}$. This fraction represents
poorly crystalline Fe(III) oxides. Particulate Mn, hereafter $Mn_{(DCA)}$ was extracted with
dithionite-citrate-acetic acid (DCA; pH 4.8) for 4 h from air-dried sediment and was
determined by inductive coupled plasma-atomic emission spectrometry (ICP-AES, Optima
3300DV, Perkin-Elmer Co). The DCA extraction aims at dissolving free Mn oxides and
authigenic Mn(II) phases. The reproducibility of the measurements was better than 10 % and
the detection limits was 3 μM for Mn. For the determination of total reduced sulfur (TRS)
that includes acid volatile sulfide (AVS = FeS + $H_2S$) and chromium-reducible sulfur (CRS =
$S^0$ + $FeS_2$), sediment samples were fixed with Zn acetate, and sulfide was determined
according to the method of Cline (1969) after a two-step distillation with cold 12 M HCl and





boiling 0.5 M $Cr^{2+}$ solution (Fossing and Jørgensen, 1989). The contents of particulate
organic carbon (POC) and nitrogen (PON) in the surface sediment were analyzed using a
CHN analyzer (CE Instrument, EA 1110) after removing $CaCO_3$ using 12 M HCl.

### 2.6 Oxygen micro-profiles

Oxygen profiles were measured at 50 µm resolution using Clark-type microelectrodes
(Unisense, OX-50) while stirring the overlying water. Microelectrodes were calibrated
between 100 % air-saturated *in situ* bottom water and $N_2$ purged anoxic bottom water. Three
profiles were measured at each site. The diffusive boundary layer (DBL) and sediment-water
interface (SWI) were determined according to Jørgensen and Revsbech (1985). To estimate
the volume-specific oxygen consumption rate, we used the PROFILE software (Berg et al.,

1998).

### 2.7 Rate measurements

The diffusive oxygen uptake (DOU) was calculated from the calibrated oxygen microprofiles.
$$DOU = - D_o\, \Delta C/\Delta z, \tag{1}$$
where $D_o$ is the temperature-corrected molecular diffusion coefficient, and C is the oxygen
concentration at depth z within the diffusive boundary layer (DBL) (Jørgensen and Revsbech,

1985).

The volume-specific $O_2$ consumption rates exhibited a bimodal depth distribution with
activity peaks near the sediment-water interface and the oxic/anoxic interface, respectively.
Thus, $O_2$ consumption rates by aerobic organotrophic respiration was defined as the $O_2$
consumption rate near the sediment-water interface, whereas the oxygen consumption at the
oxic-anoxic interface was assigned to re-oxidation of reduced inorganic compounds
(Rasmussen and Jørgensen, 1992; Canfield et al., 2005).
Total anaerobic $C_{org}$ mineralization rates were determined by linear regression of the
accumulation of total DIC with time during the anoxic bag incubations (Fig. 3) after
correcting for $CaCO_3$ precipitation (Thamdrup et al., 2000). Briefly, $CaCO_3$ precipitation was
calculated from decreasing soluble $Ca^{2+}$ concentration during the anoxic bag incubation:




$\Delta CaCO_3 = \Delta[Ca^{2+}]_{sol} \times (1 + K_{Ca})$,                                           (2)

where, $K_{Ca}$ is the adsorption constant for $Ca^{2+}$ ($K_{Ca} = 1.6$) (Li and Gregory, 1974). Then rate
of DIC production rate corrected for $CaCO_3$ precipitation was calculated as:

DIC production = DIC accumulation + $CaCO_3$ precipitation                        (3)

Fe(III) reduction rates were determined by linear regression of the increase in solid-phase

$Fe(II)_{(oxal)}$ concentration with time during anoxic bag incubations. The dissimilatory microbial
Fe(III) reduction rate was derived by subtracting abiotic Fe reduction coupled to the
oxidation of sulfide produced by sulfate reduction (Kostka et al., 2002b):

Dissimilatory microbial Fe(III) Red = Total Fe(III) Red – Abiotic Fe(III) Red           (4)

assuming that abiotic Fe reduction coupled to $H_2S$ oxidation occurred at a stoichiometry of 2
Fe(III) per 3 $H_2S$ (Pyzik and Sommer, 1981):

$3H_2S + 2FeOOH = 2FeS + S^o + 4H_2O$                                           (5)

Finally, to estimate the $C_{org}$ oxidation by microbial Fe reduction, the 4:1 stoichiometry of

iron reduction coupled to $C_{org}$ oxidation was used from the stoichiometric equation (Canfield
et al., 1993a):

$CH_2O + 4FeOOH + 8H^+ = CO_2 + 4Fe^{2+} + 7H_2O$                              (6)


Mn reduction rates were determined from linear regression of the production of dissolved

$Mn^{2+}$ with time during the anoxic bag incubations. Similar to previous studies (e.g., Canfield
et al., 1993a, 1993b; Thamdrup and Dalsgaard, 2000), we assumed that accumulating
dissolved Mn was $Mn^{2+}$. This ignores a potential contribution from $Mn^{3+}$, which in some
cases can constitute a substantial fraction of the dissolved Mn pool at the upper boundary of
the zone with soluble Mn accumulation in marine sediments (Madison et al., 2013). Further
studies of the dynamics of soluble $Mn^{3+}$ are required to evaluate its potential importance in



anoxic incubations. Such studies pending, we find justification for our assumption in the
good agreement observed in the previous studies between Mn reduction rates calculated
based on the assumption that soluble Mn is $Mn^{2+}$ (Eq. 7) and independent estimates of rates
of carbon mineralization through dissimilatory Mn reduction based on DIC or $NH_4^+$
accumulation. Due to strong adsorption of $Mn^{2+}$ to Mn oxide surfaces, (Canfield et al.,
1993b), the Mn reduction rates were estimated after compensating for the adsorption effect of
$Mn^{2+}$ to Mn-oxides according to Thamdrup and Dalsgaard (2000):

Mn reduction rate = $Mn^{2+}$ accumulation rate $\times (1 + K^*_{Mn}{}^{2+} \times (1 - \Phi) \times \Phi^{-1} \times \delta_s)$  (7)

where, $\Phi$ = porosity
$\delta_s$ = density of sediment
$K^*_{Mn}{}^{2+} = 4.8 + 0.14 \times [Mn(IV)]$ (ml $g^{-1}$)
$[Mn(IV)]$ = the concentration of Mn(IV) ($\mu$mol $g^{-1}$)

We here assume that extracted $Mn_{(DCA)}$ represents Mn(IV) as observed in surface
sediments of another Mn-rich site (Canfield et al., 1993b, Thamdrup and Dalsgaard, 2000).
Small levels of $Mn_{(DCA)}$ remaining at depth further suggest that little Mn(II) accumulates in
the solid phase (*see* Results). $C_{org}$ oxidation by dissimilatory Mn(IV) reduction was
calculated from the stoichiometric equation (Canfield et al., 1993a):

$CH_2O + 2MnO_2 + 4H^+ = CO_2 + 2Mn^{2+} + 3H_2O$  (8)

Sulfate reduction rates were determined using the radiotracer method of Jørgensen (1978).
Sediment cores (35 cm long with 2.9 cm i.d.) were collected in triplicate, injected
horizontally at 1-cm vertical interval with 5 $\mu$L radiolabeled sulfate ($^{35}S$-$SO_4^{2-}$, 15 kBq $\mu l^{-1}$,
Amersham) diluted in sterilized NaCl solution (3.0 %), and incubated for 12 h at *in situ*
temperature. At the end of the incubation, the sediment was sliced into sections, fixed in Zn
acetate (20 %), and frozen at −25 °C until processed in the laboratory. The reduced $^{35}S$ was
recovered using distillation with a boiling acidic $Cr^{2+}$ solution according to Fossing and
Jørgensen (1989). Background radioactivity of $^{35}S$ was 32.4±3.7 cpm $cm^{-3}$ (*n*=10) at site D3
and 87.5±38.7 cpm $cm^{-3}$ (*n*=10) at site M1. Detection limits of the SRR, estimated from the



double standard deviation of the blank value (i.e., 7.4 and 77.4 cpm) according to Fossing et
al. (2000), ranged from 0.79 to 2.62 nmol cm$^{-3}$ d$^{-1}$. To elucidate the contribution of sulfate
reduction in anaerobic carbon oxidation, the SRRs (Fig. 5B, 5G) were converted to carbon
oxidation using a stoichiometric equation (Thamdrup and Canfield, 1996):

$2CH_2O + SO_4^{2-} + 2H^+ = 2CO_2 + H_2S + 2H_2O$ $\qquad\qquad\qquad$ (9)


**3    Results**

**3.1    Pore-water and solid-phase constituents**

The depth distributions of $NH_4^+$, $NO_3^-$, $Mn^{2+}$ and $Fe^{2+}$ in the pore-water as well as solid phase
Mn, Fe and S for the two stations are shown in Fig. 2. $NH_4^+$ concentrations at M1 increased
steadily with depth (Fig. 2A) whereas at D3 it decreased down to 3 cm depth before it
increased below (Fig. 2F). Highest concentrations of nitrate were measured at 0 to 1 cm
sediment depth at the two stations and concentrations decreased below a background level (<
2 µM) below 1 cm at both M1 and D3 (Fig. 2A, 2F). Dissolved $Mn^{2+}$ concentrations differed
widely between the sites showing a maximum of 56 µM between 0 and 3 cm depth and not
exceeding 10 µM below at M1 (Fig. 2B), whereas at D3 concentrations increased to a
maximum of 286 µM at 10 – 12 cm depth (Fig. 2G). Conversely, dissolved $Fe^{2+}$
concentrations at M1 increased from 11 µM at 0 – 0.5 cm to 32 µM at 6 – 7 cm depth, and
stayed constant below (Fig. 2C), whereas at D3, concentrations were uniformly low showing
a slight increase to 12 µM at 15 cm (Fig. 2F).

Extractable Mn (Mn$_{(DCA)}$) concentrations were low ( < 3 µmol cm$^{-3}$) in the upper 20 cm at

the slope site (M1) (Fig. 2B), but up to 200 µmol cm$^{-3}$ in the upper 4 cm depth of the
sediment at the center of the basin (D3), with a sharp decrease to near depletion ( ~1 µmol
cm$^{-3}$) below 10 cm (Fig. 2G). At the slope site (M1), concentrations of Fe(III)$_{(oxal)}$ decreased
slightly with increasing depth from 28 µmol cm$^{-3}$ near the surface to 13 µmol cm$^{-3}$ at 20 cm
depth, mirroring an increase in Fe(II)$_{(oxal)}$ (Fig. 2D). At the center of the basin (D3),
Fe(III)$_{(oxal)}$ increased slightly from 67 µmol cm$^{-3}$ at 0 – 0.5 cm to 90 µmol cm$^{-3}$ at 4 – 6 cm
depth, and decreased steeply below to 4.8 µmol cm$^{-3}$ at 12 – 14 cm depth (Fig. 2I). Of total



Fe$_{(oxal)}$, Fe(III)$_{(oxal)}$ comprised > 98 % at 0 – 2 cm and > 97% at 0 – 8 cm depth at M1 and D3,
respectively. The fraction of Fe(III)$_{(oxal)}$ in Fe$_{(oxal)}$ then decreased to 40 % at 10 – 12 cm depth
at both sites. Acid volatile sulfur (AVS) exhibited a slight increase with depth at M1 from 0.8
µmol cm$^{-3}$ at the surface to 7.2 µmol cm$^{-3}$ at 20 cm depth (Fig. 2E), but was not detected at
D3 (Fig. 2J). Concentrations of chromium reducible sulfur (CRS) at M1 increased rapidly
with depth from 1.9 µmol cm$^{-3}$ at 0 – 0.5 cm to 21.8 µmol cm$^{-3}$ at 20 cm depth (Fig. 2D),
whereas the CRS concentration remained < 0.1 µmol cm$^{-3}$ at D3 (Fig. 2J).

**3.2     O$_2$ microprofiles and diffusive oxygen utilization rate**

Oxygen penetrated less than 4 mm into the sediments (Fig. 3), and rates of diffusive oxygen
utilization (DOU) were 7.1 and 6.0 mmol O$_2$ m$^{-2}$ d$^{-1}$ at M1 and D3, respectively (Table 2).
Oxygen consumption by aerobic respiration estimated from the O$_2$ micro-profiles (area I and
II in Fig. 3) was higher at the slope site M1 (4.0 mmol O$_2$ m$^{-2}$ d$^{-1}$) than at the D3 in the center
of the basin (2.5 mmol O$_2$ m$^{-2}$ d$^{-1}$). O$_2$ consumption by re-oxidation of reduced inorganic
compounds indicated by increased activity at the oxic/anoxic interface (area III in Fig. 3)
accounted for 43 % and 57 % of the DOU at M1 and D3, respectively. From the profiles of
geochemical constituents (Fig. 2), O$_2$ consumption was mainly attributed to the re-oxidation
of sulfide and Fe$^{2+}$ at M1 and of Mn$^{2+}$ at D3.

**3.3     Anoxic bag incubations**

Changes in concentrations of DIC, Ca$^{2+}$, dissolved Mn$^{2+}$ and solid Fe(II)$_{(oxal)}$ over time during
anoxic bag incubations from sediment of 0 – 2, 2 – 4, 4 – 6 and 6 – 8 cm depth intervals are
presented in Fig. 4. The DIC concentrations increased linearly over time during incubations
of sediment in all bags from M1 and D3, except the bag from 6 – 8 cm at D3. The DIC
accumulation rates were generally higher at the slope site (M1) than at the basin site (D3)
(Table 4). The concentrations of Ca$^{2+}$ decreased with time at all depth intervals of M1,
whereas a decrease of Ca$^{2+}$ was observed only for the 2 – 4 cm depth interval at D3. The
decrease of Ca$^{2+}$ indicates CaCO$_3$ precipitation, which consequently underestimates DIC
accumulation, especially at M1.
Coinciding with high concentrations of solid Mn$_{(DCA)}$ (Fig. 2G), prominent Mn$^{2+}$
accumulation appeared at 0 – 6 cm depth of D3, whereas no increase of Mn$^{2+}$ was observed at





M1 except a slight accumulation at $0 – 2$ cm interval (Fig. 4). Solid $Fe(II)_{(oxal)}$ concentrations
increased linearly with time at $0 – 4$ cm depth of M1, whereas highest $Fe(II)_{(oxal)}$
accumulation was observed at $4 – 6$ cm depth at D3. An increase of $Fe(II)_{(oxal)}$ was not
discernible in the Mn-oxide-rich surface sediment ($0 – 2$ cm) of D3.

**3.4 Sulfate reduction rates (SRR)**

At the slope site (M1), SRR increased from 18 nmol $cm^{-3}$ $d^{-1}$ at the surface to $97 – 103$ nmol
$cm^{-3}$ $d^{-1}$ at $1.5 – 2$ cm depth, and decreased below to 12.5 nmol $cm^{-3}$ $d^{-1}$ at 20 cm depth (Fig.
5B). In contrast, SRR at the manganese oxide-rich basin site (D3) ranged from 1.7 to 8.7
nmol $cm^{-3}$ $d^{-1}$, and did not vary with depth. Depth integrated SRR down to 10 cm depth was
10 times higher at M1 (4.3 mmol $m^{-2}$ $d^{-1}$) than at D3 (0.4 mmol $m^{-2}$ $d^{-1}$) (Table 3).

**3.5 DIC production rates**

Vertical profiles of the DIC production rate, that were derived from the linear regression of
the DIC production measured in anoxic bag incubation (Fig. 4) after correcting for $CaCO_3$
precipitation, are presented in Fig. 5C and 5H for M1 and D3, respectively. At M1, the DIC
production rates decreased with depth from 280 nmol $cm^{-3}$ $d^{-1}$ ($0 – 2$ cm depth) to 69 nmol
$cm^{-3}$ $d^{-1}$ ($8 – 10$ cm depth) (Fig. 5C), whereas the DIC production rates at D3 were relatively
similar across the upper 6 cm ranging from 86 to 136 nmol $cm^{-3}$ $d^{-1}$, and decreased to $8 – 15$
nmol $cm^{-3}$ $d^{-1}$ at $6 – 10$ cm (Fig. 5H). The integrated DIC production rate within 10 cm depth
of the sediment was twice as high at M1 (14.0 mmol $m^{-2}$ $d^{-1}$) as at the D3 (7.2 mmol $m^{-2}$ $d^{-1}$)
(Table 4).

**3.6 Rates of Mn and Fe reduction**

The accumulation of $Mn^{2+}$ evidenced that manganese reduction was occurring in the surface
sediment ($0 – 6$ cm) of D3 (Fig. 4). The manganese reduction rate (MnRR) derived from
$Mn^{2+}$ accumulation with correction for adsorption ranged from 7.5 nmol $cm^{-3}$ $d^{-1}$ ($0 – 2$ cm
depth) to 198 nmol $cm^{-3}$ $d^{-1}$ ($2 – 4$ cm depth) at D3 (Fig. 5I). In contrast, MnRR at M1 was
indiscernible except for low activity (2.2 nmol $cm^{-3}$ $d^{-1}$) at $0 – 2$ cm depth (Fig. 5D). Depth
integrated MnRR at D3 (8.21 mmol $m^{-2}$ $d^{-1}$) was 200 times higher than the MnRR at M1




(0.04 mmol m$^{-2}$ d$^{-1}$) (Table 3). The iron reduction rate (FeRR), derived from Fe(II)$_{(oxal)}$
accumulation, at M1 was highest in the 0 – 2 cm interval (237 nmol cm$^{-3}$ d$^{-1}$), and then
decreased with depth to 38 nmol cm$^{-3}$ d$^{-1}$ at 8 – 10 cm depth (Fig. 5E). In contrast, Fe
reduction was not detected in the surface sediment at D3, but increased to its maximum rate
of 240 nmol cm$^{-3}$ d$^{-1}$ at 4 – 6 cm depth. The FeRR then decreased with depth to 12 nmol cm$^{-3}$
d$^{-1}$ at 8 – 10 cm (Fig. 5J) where a few data points were adopted to derive the line of best-fit
regression. Depth integrated total FeRR was slightly higher at M1 (11.4 mmol m$^{-2}$ d$^{-1}$) than at
D3 (7.53 mmol m$^{-2}$ d$^{-1}$) (Table 3). The ratio of microbial Fe reduction, Fe Red$_{(microbial)}$, to
abiotic Fe reduction coupled to sulfide oxidation, Fe Red$_{(abiotic)}$, ranged from 1.14 (8 – 10 cm
at M1) to 52.3 (2 – 4 cm at D3), which indicated that the Fe reduction at Mn- and Fe oxides
rich basin site was mostly a microbiologically mediated process (Table 3).


**4 Discussion**

**4.1 Partitioning of C$_{org}$ oxidation in accordance with the distribution of terminal**
**electron acceptors**

One of the most prominent features revealed from the vertical distributions of geochemical
constituents at the basin site (D3) was that electron acceptors such as O$_2$, nitrate, Mn- and Fe
oxides were systematically zonated with discrete sequential depletion according to the order
of decreasing energy yield for C$_{org}$ oxidation (Fig. 5F). Such biogeochemical zones are not
sharply separated in most aquatic sediments due to, e.g., sediment heterogeneity and mixing
resulting from bioirrigation, bioturbation, and bottom turbidity currents. The profiles of
dissolved and solid phase geochemical constituents in the sediment provide indications as to
specific diagenetic reactions prevailing (Froelich et al., 1979). However, reoxidation of
reduced inorganic compounds often mask the primary reactions involved in carbon oxidation
(Sørensen and Jørgensen, 1987, Hines et al., 1991). Together with the discrete geochemical
zonation of the electron acceptors, the independently executed metabolic rate measurements
(Fig. 5) allowed us to evaluate the relative contribution of each terminal electron-accepting
pathway with sediment depth.
Previous experimental studies that have quantified pathways of anaerobic carbon
oxidation in subtidal marine sediments have generally determined the contributions of Mn




and Fe reduction indirectly from the difference between rates of DIC production and sulfate
reduction converted to carbon equivalents (e.g., Canfield et al., 1993b; Thamdrup and
Canfield, 1996; Vandieken et al., 2006). The inferred rates of Mn and Fe reduction were
further supported by the depth distribution of metal oxides and patterns of $Mn^{2+}$ and $Fe^{2+}$
accumulation in the pore water, but could not be verified because the accumulation of
particulate Mn(II) and Fe(II) – which represents the overwhelming fraction of the reduced
pools – was not quantified. Here, we combined the indirect approach with independent
determination of Mn and Fe reduction rates. Thus, we obtained two separate estimates of
anaerobic carbon oxidation rates; based on DIC production and on the sum of sulfate, Fe, and
Mn reduction converted to carbon equivalents, respectively (Table 4). At M1, within the 0 –
10 cm depth interval, the average ratio between total anaerobic $C_{org}$ oxidation rate (10.7
mmol C $m^{-2}$ $d^{-1}$) and the $C_{org}$ oxidation from DIC production (14.0 mmol C $m^{-2}$ $d^{-1}$) was 0.77
(Table 4). Similarly, at D3, the average ratio between total anaerobic $C_{org}$ oxidation (6.79
mmol $m^{-2}$ $d^{-1}$) and anaerobic DIC production (7.22 mmol $m^{-2}$ $d^{-1}$) was 0.94. Consequently,
there was a good agreement between the two estimates with a ratio of total anaerobic $C_{org}$
oxidation by Mn + Fe + sulfate : DIC production for individual depth intervals of 0.8 – 1.2
(Table 4) with the exception at the 0 – 2 cm depth of slope site (M1) where the ratio was
slightly lower, 0.66, possibly due to a contribution from the $C_{org}$ oxidation by nitrate
reduction. The similarity of the two estimates across all incubations spanning a range of
redox conditions provides confidence in our approach for calculating dissimilatory Mn and
Fe reduction rates. Specifically, the good agreement indicates that the underlying
assumptions concerning Mn adsorption and reactions of Fe(III) and sulfide are valid as first-
order approximations. The general agreement further supports the validity of previous
determinations of dissimilatory Mn and Fe reduction rates based on the difference between
DIC production and $SO_4^{2-}$ reduction. (Canfield et al., 1993a, 1993b; Thamdrup et al., 2000;
Vandieken et al., 2006; Vandieken et al., 2014).
To elucidate the contribution of sulfate reduction in anaerobic carbon oxidation, the SRRs
(Fig. 5B, 5G) were converted to carbon oxidation (Thamdrup and Canfield, 1996), and then
compared to the DIC production rates from anoxic bag incubation (Fig. 5C, 5H). At the slope
site (M1), the fraction of anaerobic $C_{org}$ oxidation coupled to sulfate reduction increased with
depth from 48 % at 0 – 2 cm, to > 90 % at 8 – 10 cm (Table 4). Thus, the excess $C_{org}$
oxidation in the upper layers should be coupled to other electron accepting processes. Indeed,
the $C_{org}$ oxidation by Fe reduction (0.96 mmol $m^{-2}$ $d^{-1}$) accounted for most of the remaining



anaerobic $C_{org}$ oxidation (11 – 18 % of DIC production) at 0 – 8 cm depth, consistent with the
distribution of Fe(III) decreasing from > 25 µmol cm$^{-3}$ near the surface (Fig. 6, Table 4). Mn
reduction was of minor importance at M1 because of the low content of Mn oxide (< 3 umol
cm$^{-3}$). Carbon oxidation coupled to aerobic respiration was estimated to 3.1 mmol m$^{-2}$ d$^{-1}$
corresponding to 18 % of the total aerobic + anaerobic oxidation, while the contributions of
Fe and sulfate reduction to this total were 12 % and 50 %, respectively (Table 4). As
mentioned above, nitrate reduction/denitrification may contribute part of the unexplained 20 %
of carbon oxidation, but most of this imbalance likely reflects the combined uncertainties in
the estimates of the individual pathways. As discussed further below, the partitioning of $C_{org}$
oxidation at M1 falls within the range previously reported for continental margin sediments.

In contrast to M1, $C_{org}$ oxidation by sulfate reduction at the basin site (D3) accounted for

only a small fraction (<10 %) of anaerobic $C_{org}$ oxidation at 0 – 6 cm interval and it only
dominated carbon oxidation at 8 – 10 cm (Fig. 5H, Table 4). Oxygen and $NO_3^-$ were depleted
within 3.6 mm and 1 cm depth of the sediment surface, respectively (Fig. 5F), while Mn and
Fe(III) oxides were abundant at 0 – 4 cm and 0 – 6 cm, respectively. Consistent with the
abundance of electron acceptors, high rates of Mn and Fe reduction (Fig. 5I and 5J) implied
Mn and Fe reduction as the most significant $C_{org}$ oxidation pathways to 6 cm depth. At 0 – 2
cm depth, $C_{org}$ oxidation by aerobic respiration and Mn reduction accounted for 53 % and 43 %
of total $C_{org}$ oxidation, respectively (Fig. 6). At 2 – 4 cm, Mn reduction accounted for 73 % of
total $C_{org}$ oxidation and 92 % of anaerobic $C_{org}$ oxidation (Table 4, Fig. 6). Its importance
decreased to 22 % at 4 – 6 cm due to lower Mn concentrations, while microbial Fe(III)
reduction here contributed 51 %, and the partitioning of sulfate reduction increased to 11 %
(Fig. 6). Consequently, the relative distribution of each $C_{org}$ oxidation pathway with depth at
D3 (Fig. 6) matched well with the depth distribution of respective electron acceptors (Fig. 5F).
Overall, within the 10 cm depth sediments intervals, Mn and Fe reduction were the dominant
$C_{org}$ oxidation pathways comprising 45 % and 20 % of total carbon oxidation, respectively, at
the Mn and Fe oxide-rich site in the center of the UB (Table 4).

Despite the high Fe oxide content at 0 – 4 cm at D3 (Fig. 5F), no solid $Fe(II)_{(oxal)}$

accumulation was observed at this depth range (Fig. 6). This indicates that Fe(III) reduction
may not occur under this Mn-oxide rich conditions. Indeed, using a combination of 16S
rRNA-stable isotope probing and geochemical analysis in three manganese oxides-rich
sediments including the UB, Vandieken et al. (2012) identified bacteria related to *Colwellia*,
*Oceanospillaceae* and *Arcobacter* as acetate-oxidizing bacteria that potentially reduce



manganese, whereas no known iron reducers were detected in the Mn-rich sediment.
Similarly, Thamdrup et al. (2000), in Mn oxide- rich Black Sea sediment, found that the
abundance of viable Fe-reducing bacteria in most probable number counts was low in
comparison to Mn reducers and the addition of ferrihydrite did not stimulate Fe reduction,
which implied that Fe reduction should be outcompeted by the Mn reduction process.

Nonetheless, Mn reduction estimated from the increase of $Mn^{2+}$ at $0 - 4$ cm interval at D3
(Fig. 6) could be due to oxidation of $Fe^{2+}$ or sulfide. $Fe^{2+}$ may readily react with Mn oxides
(Myers and Nealson, 1988; Lovley and Phillips, 1988) by the reaction $2Fe^{2+} + MnO_2 + 4H_2O$
$= Mn^{2+} + 2Fe(OH)_3 + 2H^+$. However, in the Mn oxide-rich sediment of the Skagerrak,
Canfield et al. (1993b) revealed that the addition of Ferrozine, a strong complexation agent
for $Fe^{2+}$, had no inhibitory effect on the $Mn^{2+}$ liberation, indicating that the chemical reaction
of $MnO_2$ with $Fe^{2+}$ generated by Fe reduction was not responsible for the accumulation of
$Mn^{2+}$. As manganese reduction is thermodynamically more favorable than iron and sulfate
reduction, the $Mn^{2+}$ liberation (Fig. 3) likely resulted from dissimilatory Mn reduction.

Despite anoxia and nitrate depletion, Mn reduction rates at $0 - 2$ cm depth (Fig. 5I) based
on $Mn^{2+}$ accumulation were substantially lower than the rates inferred from DIC
accumulation (Fig. 5H). A similar discrepancy was previously observed for the uppermost
part of the Mn reduction zone (Thamdrup et al., 2000), and is likely explained by particularly
strong sorption of $Mn^{2+}$ to fresh Mn oxide surfaces, which is not included in the adsorption
coefficient used here. Previous estimation of denitrification in $0 - 2$ cm depth of the UB
ranged from 0.01 to 0.17 mmol N m$^{-2}$ d$^{-1}$ (Lee, 2009), which is equivalent to a $C_{org}$ oxidation
of $0.013 - 0.213$ mmol C m$^{-2}$ d$^{-1}$ using the stoichiometric equation of $4H^+ + 5CH_2O + 4NO_3^-$
$= 5CO_2 + 2N_2 + 7H_2O$. Based on the average, the contribution of carbon oxidation by
denitrification (0.11 mmol C m$^{-2}$ d$^{-1}$) should be minor at the basin site ($\leq$ 3 % of total $C_{org}$
oxidation at $0 - 2$ cm; ~1 % of integrated $C_{org}$ oxidation). This is consistent with the general
consensus that $C_{org}$ oxidation by denitrification is of little importance in most marine
sediments (Sørensen et al., 1979; Canfield et al., 1993a; Trimmer and Engström, 2011).
Denitrification may be even further suppressed in Mn-rich sediments due to competitive
inhibition from Mn reduction (Trimmer et al., 2013).

### 4.2   $C_{org}$ oxidation dominated by manganese reduction in the UB


Microbial Fe reduction has been quantified directly in sediments of various coastal oceans



(Gribsholt et al., 2003; Kostka et al., 2002a, 2002b; Hyun et al., 2007, 2009b) and indirectly
in deeper continental margins (Thamdrup and Canfield, 1996; Jensen et al., 2003; Kostka et
al., 1999). Earlier estimation from 16 different continental margin sediments indicated that
Fe(III) reduction contributed 22 % on average to anaerobic carbon oxidation (Thamdrup,
2000). Thus, the contributions from Fe(III) reduction of 12 % and 20 % of anaerobic $C_{org}$
oxidation on the slope (M1) and in the basin (D3) of the UB (Table 4) falls in the range of the
previous indirect estimates.

Unlike Fe reduction, direct estimation of manganese reduction rates is not easy, mainly

because of the restriction of the process to a thin surface layer (Sundby and Silverberg, 1985),
the rapid reduction of manganese oxides with $H_2S$ and $Fe^{2+}$ (Postma, 1985; Burdige and
Nealson, 1986; Kostka et al., 1995; Lovley and Phillips, 1988), and the adsorption of $Mn^{2+}$ to
Mn oxide surface (Canfield et al., 1993b). For that reason, only two studies, from the
Skagerrak and Black Sea, are available for direct comparison on the partitioning of Mn
reduction. The process has also been indicated to be of importance in the Panama Basin based
on diagenetic modeling (Aller, 1990) and at some Artic shelf sites where it was however not
quantified separately from Fe reduction (Vandieken et al., 2006, Nickel et al., 2008). Mn
reduction was responsible for over 90 % of total $C_{org}$ oxidation at 600 m depth in the
Skagerrak, and accounted for 13 – 45 % of anaerobic $C_{org}$ oxidation in the Black Sea shelf
sites at 60 – 130 m of water depth. To our knowledge, this report of $C_{org}$ oxidation dominated
by Mn reduction comprising 45 % of total $C_{org}$ oxidation and 57 % of anaerobic $C_{org}$
respiration in the center of the UB (Table 4) represents the first from deep-offshore basin of
the eastern Asian marginal seas.

The difference in partitioning of Mn reduction in $C_{org}$ oxidation between the UB, Black

Sea and Skagerrak reflects the close relationship between Mn oxide content in the sediment
and Mn reduction (Thamdrup et al., 2000). From the vertical distribution of electron
acceptors (Fig. 5J) and contribution of each $C_{org}$ oxidation pathway with depth (Fig. 6), it is
apparent that the availability of Mn(IV) largely controls the relative contribution to C
oxidation. In the Skagerrak, the Mn oxides are abundant in high concentration down to 10 cm
depth (Canfield et al., 1993b), whereas Mn oxides in the Black Sea and the Ulleung Basin
were enriched only down to 2 cm and 4 cm, respectively (Thamdrup et al., 2000, Fig. 2).
Using the available data set for the three marine sediments, we further plotted the relative
contribution of manganese reduction to total carbon oxidation as a function of Mn-oxides
concentration to expand data from Thamdrup et al., 2000 (Fig. 7). The plot indicates



saturation kinetics with a close correlation between Mn oxide content and the importance of
Mn reduction at low concentrations. Curve-fitting yields a concentration of $MnO_2$ at 50 % of
contribution of manganese reduction to total $C_{org}$ oxidation ($K_s$) of 8.6 µmol cm$^{-3}$ similar to
the approx. 10 µmol cm$^{-3}$ suggested before (Thamdrup et al., 2000). This indicates that Mn
reduction can be a dominant $C_{org}$ oxidation process even at low concentrations of Mn oxides
compared to those found at UB. Manganese enrichments of this magnitude have been
reported for several locations on the continental margins (Murray et al., 1984; Gobeil et al.,
1997; Haese et al., 2000; Mouret et al., 2009; Magen et al., 2011; Macdonald and Gobeil,
2012) in addition to the relatively few places where dissimilatory Mn reduction was already
indicated to be of importance, as discussed above. Thus, the process may be of more
widespread significance on continental margins.

**4.3   Source of high Mn oxide content**

The strong enrichment of Mn in the UB surface sediment is primarily of diagenetic origin as
indicated by similar Mn concentrations at depth in the sediment at D3 (0.95 – 3.02 µmol cm$^{-3}$)
compared to M1 (0.36 – 3.74 µmol cm$^{-3}$) (Fig. 2) combined with higher sediment
accumulation rates at the slope (0.15 – 0.3 cm y$^{-1}$) than in the basin (0.07 cm y$^{-1}$; Cha et al.,
2005). Thus, the burial flux of Mn, and thereby the net input assuming steady state deposition,
is higher at M1 than at D3. Furthermore, Mn is likely subject to geochemical focusing in the
basin as Mn depositing at shallower depths is reductively mobilized and incompletely
oxidized in the thin oxic surface layer, resulting in release to the water column and net down-
slope transport, as inferred in other ventilated basins (Sundby and Silverberg, 1985; Canfield
at al., 1993b). A diagenetic source of Mn enrichment was also concluded in previous studies
(Yin et al., 1989; Cha et al., 2007; Choi et al., 2009). The Mn remaining and being buried at
M1 likely represents unreactive detrital forms to a larger extent than at D3 (Cha et al., 2007).
Adopting the sediment accumulation rate of 0.07 cm y$^{-1}$ in the UB determined at a station 50
km from D3 (Cha et al., 2005), the average $Mn_{(DCA)}$ concentration of 1.1 µmol cm$^{-3}$ at 10 –
20 cm depth (Fig. 2G) corresponds to a flux for permanent burial of 0.002 mmol m$^{-2}$ d$^{-1}$ or
just 0.03 % of the Mn reduction rate (Table 3), i.e., an Mn atom is recycled 3800 times before
it finally gets buried. This is a much more extensive recycling than found in the Mn sediment
of Skagerrak (130 – 260 times; Canfield et al., 1993b). The difference results mainly from a
much higher burial flux of Mn (as authigenic Mn[II]) in the Skagerrak (~40 µmol cm$^{-3}$;





Canfield et al., 1993b). The reason that little, if any, authigenic Mn(II) is buried in the UB is
not clear.
As noted in previous studies (Aller 1990, Canfield et al. 1993b), high contributions of Mn
and Fe reduction to carbon oxidation in off-shore sediments requires physical mixing, which
typically occurs through bioturbation. This is also the case for the UB, where the burial flux
from the oxic surface layer into the Mn reduction zone corresponded to 0.4 mmol m$^{-2}$ d$^{-1}$ or 5 %
of the Mn reduction rate (213 µmol cm$^{-3}$ x 0.07 cm y$^{-1}$). Bioturbation has previously been
inferred, but not quantified, from $^{210}$Pb profiles in the UB (Cha, 2002), and thin polychaete
worms were observed during our sampling. Assuming bioturbation to be a diffusive process,
we estimate, in a similar manner as in the previous studies and based on the average gradient
in Mn$_{(DCA)}$ from 0.5 – 1 to 7 – 8 cm, that the Mn reduction rate would be supported at a
biodiffusion coefficient of 9.5 cm$^2$ y$^{-1}$. This value is 3.6 times lower than the coefficient
estimated for the Skagerrak (Canfield et al., 1993b) and consistent with estimates for other
sediments with similar deposition rates (Boudreau, 1994). Thus, it is realistic that
bioturbation drives Mn cycling in the UB.


**4.4 The UB as a biogeochemical hotspot**

The SRRs measured in this study (0.43 – 4.29 mmol m$^{-2}$ d$^{-1}$) are higher than those of
measured in productive systems such as the Benguela upwelling system in the Southeast
Atlantic (Ferdelman et al., 1999; Fossing et al., 2000), and even comparable to those reported
at the continental slope of the Chilean upwelling system (2.7 – 4.8 mmol m$^{-2}$ d$^{-1}$) (Thamdrup
and Canfield, 1996) at a similar depth range of 1000 – 2500 m. The total anaerobic DIC
production rates at the slope (14.0 mmol m$^{-2}$ d$^{-1}$) and basin site (7.2 mmol m$^{-2}$ d$^{-1}$) were also
comparable to those measured at the same depth range of Chilean upwelling site (9.2 – 11.6
mmol m$^{-2}$ d$^{-1}$) (Thamdrup and Canfield, 1996). Since rates of benthic carbon oxidation are
largely controlled by the supply of organic carbon (Canfield et al., 2005), a high organic flux
reflected in the high organic content (> 2.5 %, dry wt.) in the sediment of the UB (Table 1) is
likely to explain the high metabolic activities. A similar high organic carbon content as in the
UB is rarely found in deep-sea sediment underlying oxic bottom water at depths exceeding
2000 m, except for Chilean upwelling site (Lee et al., 2008). This high organic carbon
content in the UB is mainly associated with the combination of enhanced biological
production resulting from the formation of coastal upwelling (Hyun et al., 2009a), occurrence





of an intrathermocline eddy resulting in the extraordinary subsurface chlorophyll-a maximum
(Kim et al., 2012), high organic C accumulation rates exceeding 2 g C m$^{-2}$ yr$^{-1}$ (Lee et al.,
2008), and high export production (Kim et al., 2009). In addition to the large vertical sinking
flux, the lateral transport of the organic matter along the highly productive southeastern slope
of the UB also contributes to the high organic content (Lee et al., 2015). Consequently, high
benthic mineralization resulting from the high organic content in the sediment implied that
the UB is a biogeochemical hotspot where significant turnover of organic matter and nutrient
regeneration occur. Recently, a rapid increase of sea surface temperature of 1.09 ℃ in the
East Sea over the last two decades (1982 – 2006) has been recorded, which is the fourth
highest among the 18 large marine ecosystems in the world ocean (Belkin, 2009). It is thus
important to monitor any changes in the rates and partitioning of $C_{org}$ oxidation to better
understand the biogeochemical carbon, nutrients and metal cycles associated with long-term
climatic changes in the UB, the biogeochemical hotspot of the East Sea.


**5. Conclusions**

Surface sediments of the Ulleung Basin (UB) in the far east Eurasian continent are
characterized by a high organic carbon content (> 2.5 %, dry wt.) high concentrations Fe
oxides (up to 100 µmol cm$^{-3}$), and very high concentrations of Mn oxides (> 200 µmol cm$^{-3}$).
For the first time in the Asian marginal seas, and in one of only few experimental studies of
the partitioning of $C_{org}$ oxidation pathways in deep-sea sediments in general, we show that
microbial Mn and Fe reduction are the dominant $C_{org}$ oxidation pathways, comprising 45 %
and 20 % of total $C_{org}$ oxidation, respectively. The high Mn content results from highly
efficient recycling through reoxidation with very low permant burial of authigenic Mn(II)
phases. The basin topography may ensure that any Mn$^{2+}$ escaping to the overlying water
returns to the sediment after reprecipitation. The relative importance of Mn reduction to $C_{org}$
oxidation displays saturation kinetics with respect to Mn oxide content with a low half-
saturation value (8.6 µmol cm$^{-3}$), which further implies that Mn reduction can be a dominant
$C_{org}$ oxidation process in sediments with lower $MnO_2$ content, and thereby that the process
might be more important in deep-sea sediments than previously thought. Vertical
distributions of the major terminal electron acceptors such as $O_2$, nitrate, Mn- and Fe oxides
were systematically zonated with discrete sequential depletion according to the order of




decreasing energy yield for $C_{org}$ oxidation, which are not sharply separated in most aquatic
sediments due to, e.g., sediment heterogeneity and mixing resulting from bioirrigation,
bioturbation, and bottom turbidity currents. High benthic mineralization resulting from the
high organic carbon content in the sediment implied that the UB is a biogeochemical hotspot
where significant turnover of organic matter and nutrient regeneration occur. The East Sea,
including the UB, has experienced the fastest increase in sea water temperature (1.09 $^{o}$C) for
the past two decades (1982 – 2006). If this continues, the UB sediment provides with an ideal
natural laboratory to monitor changes in the rates and partitioning of $C_{org}$ oxidation in order
to better understand the biogeochemical cycling of carbon, nutrients and metals associated
with long-term climatic changes.


**Author contribution**


J-H Hyun as first author and leader of the Korean research group designed the original
experiments and conducted most writing; S-H Kim, JS Mok, and H-Y Cho participated in
onboard research activities and analytical processes; V Vandieken participated in onboard
research and was actively involved in the discussion of the manuscript; D Lee, as project
manager of the EAST-1 program, paid the ship-time and has participated in discussion of the
results; B Thamdrup, as leader of the Danish research group, collaborated with J-H Hyun in
designing the experiments and writing and discussing the manuscript.


**Acknowledgements**

This research was a part of the projects titled Korean Long-term Marine Ecological
Researches (K-LTMER) and East Asian Seas Time Series-I (EAST-I) funded by the Korean
Ministry of Oceans and Fisheries, and was also supported by the National Research
Foundation of Korea (NRF-2012-013-2012S1A2A1A01030760) in collaboration with the
Danish Council for Independent Research and the Danish National Research Foundation
(DNRF53).



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






Table 1. Environmental settings and sediment characteristics

| Environmental parameter | M1 (Continental slope) | D3 (Center of the basin) |
|---|---|---|
| Latitude | 36° 10' N | 37° 00' N |
| Longitude | 130° 10' E | 131° 00' E |
| Water depth (m) | 1,453 | 2,154 |
| Sediment temperature ($^{o}$C) | 1.3 | 0.6 |
| Pore-water salinity (psu) | 34.2 | 34.8 |
| Water content (%) | 85 (± 3.1) | 77 (± 1.8) |
| Porosity | 0.95 (± 0.03) | 0.86 (± 0.01) |
| Density (g cm$^{-3}$) | 1.10 (± 0.02) | 1.12 (± 0.02) |
| Total organic carbon (%, dry wt.) | 3.96 (± 0.27) | 2.66 (± 0.09) |
| Total organic nitrogen (%, dry wt.) | 0.38 (± 0.01) | 0.35 (± 0.01) |

Numbers in parenthesis indicate ± 1SD of triplicate samples.




Table 2. Oxygen penetration depth (OPD), diffusive oxygen utilization (DOU) rate and $O_2$ consumption
rate by aerobic respiration and re-oxidation of reduced inorganic compounds (RIC) in the pore water.

| Station | OPD (mm) | DOU (mmol $O_2$ m$^{-2}$ d$^{-1}$) | $O_2$ consumption (mmol $O_2$ m$^{-2}$ d$^{-1}$) by | |
| --- | --- | --- | --- | --- |
| | | | Aerobic respiration | Re-oxidation of RIC |
| M1 | 3.2 (± 0.20) | 7.12 (± 1.36) | 4.04 (± 2.03) | 3.07 (± 0.68) |
| D3 | 3.6 (± 0.03) | 5.95 (± 0.16) | 2.53 (± 0.72) | 3.42 (± 0.58) |

Values represent averages ± 1SD ($n = 3$)



Table 3. Depth integrated rates (mmol m⁻² d⁻¹) of Mn reduction, Fe reduction, and sulfate reduction and the partitioning of abiotic and microbial Fe(III) reduction in total Fe(III) reduction with depth.

| St. | Depth Interval (cm) | SO₄²⁻ Red | Mn Red | (a)Total Fe(III) Red | Fe reduction by (a)Abiotic Fe Red | (a)Microbial Fe Red | Fe Red(Microbial) / Fe Red(Abiotic) |
|---|---|---|---|---|---|---|---|
| M1 | 0 – 2 | 1.35 | 0.04 | 4.75 | 0.90 | 3.86 | 4.28 |
|  | 2 – 4 | 1.04 | - | 3.02 | 0.70 | 2.33 | 3.33 |
|  | 4 – 6 | 0.84 | - | 1.58 | 0.56 | 1.21 | 2.16 |
|  | 6 – 8 | 0.54 | - | 1.25 | 0.36 | 0.89 | 2.47 |
|  | 8 – 10 | 0.53 | - | 0.77 | 0.36 | 0.41 | 1.14 |
|  | Sum (0-10) | 4.30 | 0.04 | 11.4 | 2.88 | 8.70 |  |
| D3 | 0 – 2 | 0.06 | (b)3.19 | - | - | - | n.a. |
|  | 2 – 4 | 0.11 | 3.96 | 1.63 | 0.07 | 1.56 | 22.3 |
|  | 4 – 6 | 0.13 | 1.05 | 4.80 | 0.09 | 4.71 | 52.3 |
|  | 6 – 8 | 0.06 | 0.01 | 0.86 | 0.04 | 0.83 | 20.8 |
|  | 8 – 10 | 0.07 | 0.00 | 0.24 | 0.05 | 0.19 | 3.80 |
|  | Sum (0-10) | 0.43 | 8.21 | 7.53 | 0.25 | 7.29 |  |

(a)Stoichiometric equations were used to evaluate the relative significance of abiotic and microbial Fe reduction:
Abiotic reduction of Fe(III) by sulfide oxidation, $3H_2S + 2FeOOH = 2FeS + S^o + 4H_2O$; Microbial Fe(III) reduction = Total Fe(III) reduction – abiotic Fe(III) reduction.
(b)back-calculated from the C oxidation by Mn reduction in the 0 – 2 cm interval in Table 5 using the stoichiometric equation, $2MnO_2 + CH_2O + H_2O = 2Mn^{2+} + HCO_3^- + 3OH^-$.
'–' indicates that the process does not occur or is regarded as negligible at the depth interval based on the OPD for aerobic respiration and geochemical profiles or anoxic bag incubations for Mn(IV) and Fe(III) reduction
'n.a.' indicates that data are not available.





Table 4. Organic carbon ($C_{org}$) oxidation (mmol C m$^{-2}$ d$^{-1}$) by each $C_{org}$ oxidation pathway, and its partitioning in total $C_{org}$ oxidation (% Total $C_{ox}$) and anaerobic $C_{org}$ oxidation (% Anaerobic $C_{org}$ ox) at each depth interval within 10 cm of the sediment. Mn Red, Mn reduction; Fe Red, Fe reduction; and $SO_4^{2-}$ Red, sulfate reduction

| St. | Depth Interval (cm) | $C_{org}$ oxidation measured by | | [c] Total $C_{org}$ oxidation (DOU + DIC) | Anaerobic $C_{org}$ oxidation by dissimilatory | | | Total anaerobic $C_{org}$ oxidation (Mn Red + Fe Red + $SO_4^{2-}$ Red) | Total Anaerobic $C_{org}$ oxidation / Anoxic DIC production |
|---|---|---|---|---|---|---|---|---|---|
| | | [a] DOU (Aerobic respiration) | [b] DIC prod. (Anaerobic respiration) | | [d] Mn Red | [d,e] Fe Red | [d] $SO_4^{2-}$ Red | | |
| M1 | 0 – 2 | 3.11 | 5.59 | 8.70 | 0.02 | 0.96 | 2.69 | 3.67 | 0.66 |
| | 2 – 4 | - | 3.31 | 3.31 | - | 0.58 | 2.09 | 2.67 | 0.81 |
| | 4 – 6 | - | 2.26 | 2.26 | - | 0.26 | 1.67 | 1.93 | 0.85 |
| | 6 – 8 | - | 1.50 | 1.50 | - | 0.22 | 1.08 | 1.30 | 0.87 |
| | 8 – 10 | - | 1.37 | 1.37 | - | 0.10 | 1.06 | 1.17 | 0.85 |
| | Sum (0 – 10) | 3.11 | 14.0 | 17.1 | 0.02 | 2.13 | 8.59 | 10.7 | 0.77 |
| | (% Total $C_{org}$ ox) | (18.1) | (81.9) | (100) | (0.13) | (12.4) | (50.1) | (62.7) | |
| | (% Anaerobic $C_{org}$ ox) | | | | (0.16) | (15.2) | (61.2) | | |
| D3 | 0 – 2 | 1.94 | 1.72 | 3.66 | [b]1.59 | - | 0.13 | 1.72 | 1.00 |
| | 2 – 4 | - | 2.72 | 2.72 | 1.98 | 0.39 | 0.22 | 2.58 | 0.95 |
| | 4 – 6 | - | 2.32 | 2.32 | 0.52 | 1.18 | 0.26 | 1.96 | 0.84 |
| | 6 – 8 | - | 0.30 | 0.30 | 0.01 | 0.21 | 0.12 | 0.33 | 1.10 |
| | 8 – 10 | - | 0.16 | 0.16 | - | 0.05 | 0.15 | 0.19 | 1.21 |
| | Sum (0 – 10) | 1.94 | 7.22 | 9.2 | 4.10 | 1.82 | 0.86 | 6.79 | 0.94 |
| | (% Total $C_{org}$ ox) | (20.6) | (78.8) | (100) | (44.8) | (19.9) | (9.41) | (77.8) | |
| | (% Anaerobic $C_{org}$ ox) | | | | (56.8) | (25.2) | (11.9) | | |

[a] Aerobic $C_{org}$ oxidation rate (= $O_2$ consumption by aerobic respiration x (106C/138$O_2$) calculated using the Redfield ratio; $O_2$ consumption by aerobic respiration rate (= DOU - re-oxidation rates) is calculated from Table 2 that is derived from the $O_2$ micro-profiles in Fig. 2.

[b] independently measured from the DIC accumulation rate in anoxic bag incubation experiment in Fig. 6 and 7.

[c] Total $C_{org}$ oxidation = aerobic $C_{org}$ oxidation + anaerobic $C_{org}$ oxidation

[d] $C_{org}$ oxidation by dissimilatory Mn(IV) reduction, Fe(III) reduction, and sulfate reduction was calculated from the stoichiometric equations: $2MnO_2 + CH_2O + H_2O = 2Mn^{2+} + HCO_3^- + 3OH^-$; $4Fe(OH)_3 + CH_2O + HCO_3^- = 4Fe^{2+} + 2HCO_3^- + 7OH^-$; $SO_4^{2-} + 2CH_2O = H_2S + 2HCO_3^-$; $H_2S = HS^- + H^+$

[e] Dissimilatory Fe(III) reduction = (Total Fe(III) reduction in Fig.7) – (Abiotic Fe(III) reduction coupled to $H_2S$ oxidation; $3H_2S + 2FeOOH = 2FeS + S° + 4H_2O$)

[f] back-calculated from: DIC production rate - (C oxidation by $SO_4^{2-}$ Red and Fe Red). See text for further discussion

' – ' indicates that the process does not occur or is regarded as negligible based on the OPD for aerobic respiration and geochemical profiles or anoxic bag incubations for Mn and Fe Red.





## Figure legends

Fig. 1. Sampling stations in the East Sea and pictures showing contrasting colors between surface sediments of the continental slope (M1) and center of the basin (D3)

Fig. 2. Concentrations of dissolved $NH_4^+$, $NO_3^-$, $Mn^{2+}$ and $Fe^{2+}$ in pore water and solid phase $Mn_{(DCA)}$, $Fe(II)_{(oxal)}$, $Fe(III)_{(oxal)}$, acid volatile sulfur (AVS) and chromium reducible sulfur (CRS) in the sediment at M1 and D3.

Fig. 3. Vertical profiles of $O_2$. The slashed area indicates the diffusive boundary layer in the sediment-water interface. The shaded area indicates that $O_2$ consumption by aerobic respiration (I and II) and re-oxidation of reduced inorganic compounds (III), respectively.

Fig. 4. Changes in pore water concentrations of DIC, $Ca^{2+}$ and $Mn^{2+}$ and solid phase $Fe(II)_{(oxal)}$ during anoxic bag incubations of sediments from 0-2, 2-4, 4-6, and 6-8 cm depth at M1 and D3. Data obtained at 8-10 cm depth interval is not shown.

Fig. 5. Vertical distribution of terminal electron acceptors ($O_2$, $NO_3^-$, Mn and Fe) and rates of sulfate reduction measured from whole core analyses, and rates of anaerobic carbon oxidation (DIC production rates), Mn reduction and Fe reduction measured from anoxic bag incubations in Fig. 4. $C_{org}$ by sulfate reduction in panel C and H was calculated from the stoichiometry of 2:1 of $C_{org}$ oxidized to sulfate reduced.

Fig. 6. Depth variations of partitioning of each carbon oxidation pathway in total carbon oxidation at M1 and D3

Fig. 7. The relative contribution of Mn reduction to total carbon oxidation as a function of the concentration of Mn(DCA) at 3 different sites. BS, Black Sea (Thamdrup et al. 2000); UB, Ulleung Basin (This study); Sk, Skagerrak (Canfield et al. 1993b).



Hyun et al. – Figure 1

Fig. 1. Sampling stations in the East Sea and pictures showing contrasting colors between surface sediments of the continental slope (M1) and center of the basin (D3)

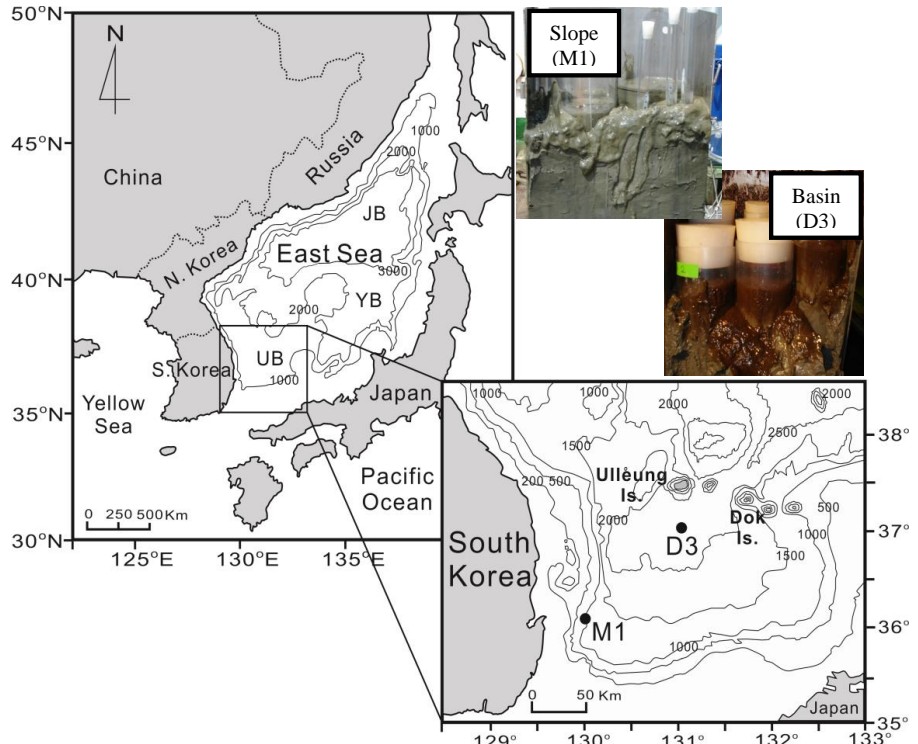





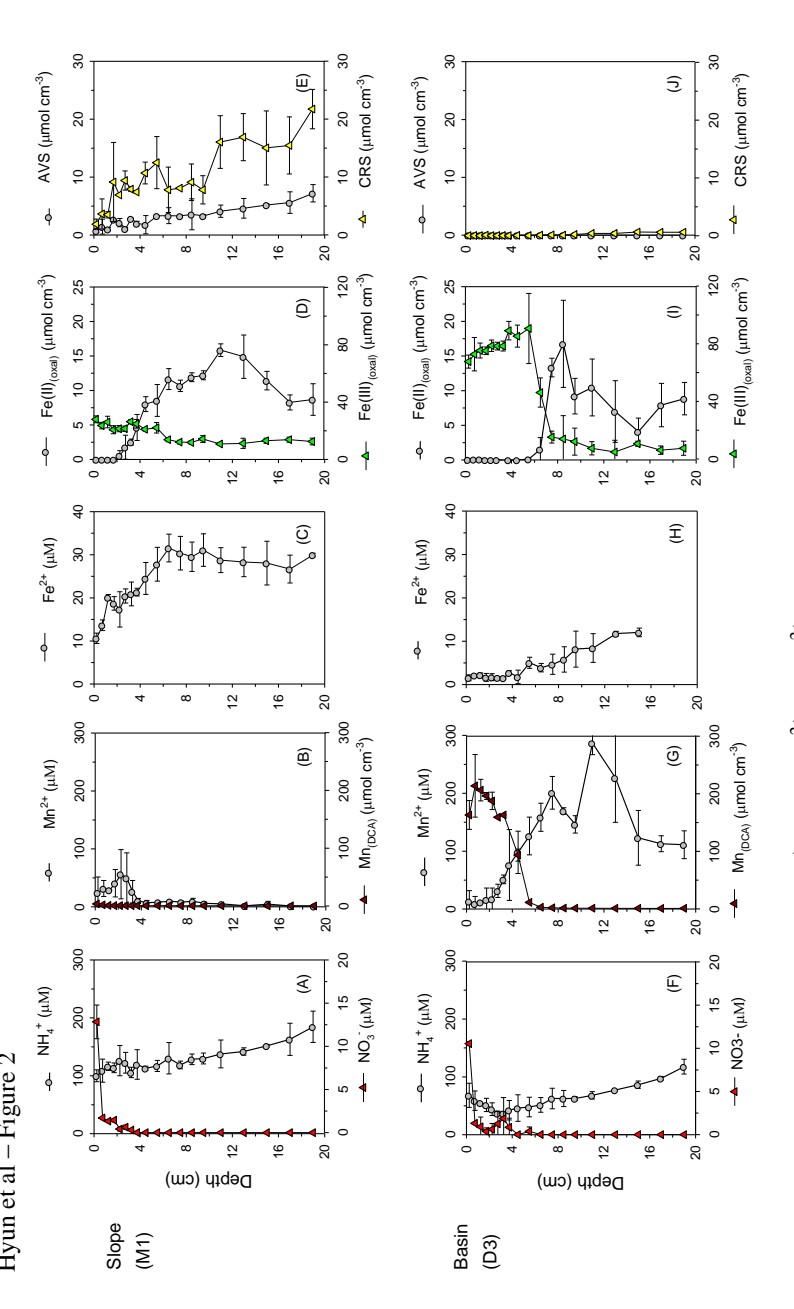

Fig. 2. Concentrations of dissolved $NH_4^+$, $NO_3^-$, $Mn^{2+}$ and $Fe^{2+}$ in pore water and solid phase $Mn_{(DCA)}$, $Fe(II)_{(oxal)}$, $Fe(III)_{(oxal)}$, acid volatile sulfur (AVS) and chromium reducible sulfur (CRS) in the sediment at M1 and D3



7    Hyun et al – Figure 3

Fig. 3. Vertical profiles of $O_2$ The slashed area indicates diffusive boundary layer in the
sediment-water interface. The shaded area indicates that $O_2$ consumption by aerobic respiration
(I and II) and re-oxidation of reduced inorganic compounds (III), respectively.





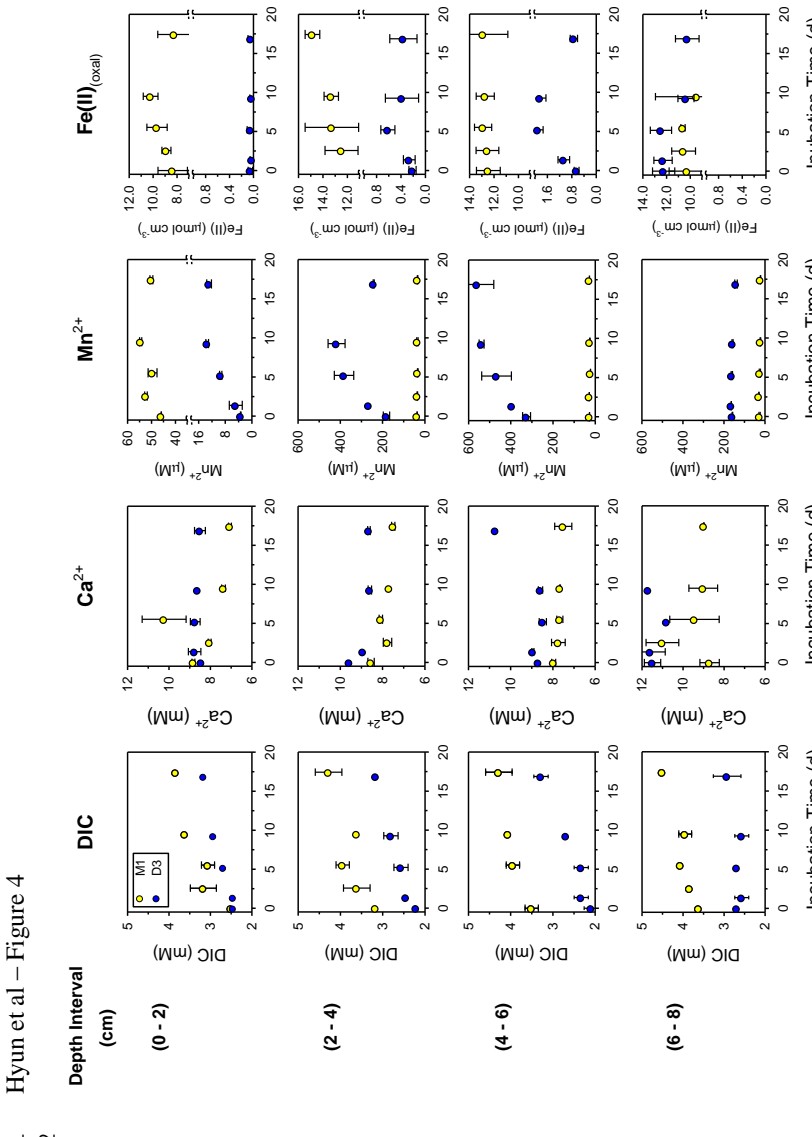

Hyun et al – Figure 4
Fig. 4. Changes in pore water concentrations of DIC, $Ca^{2+}$ and $Mn^{2+}$ and solid phase $Fe(II)_{(oxal)}$ during anoxic bag incubations of sediments
from 0-2, 2-4, 4-6, and 6-8 cm depth at M1 and D3. Data obtained at 8-10 cm depth interval is not shown.



Hyun et al. – Figure 5

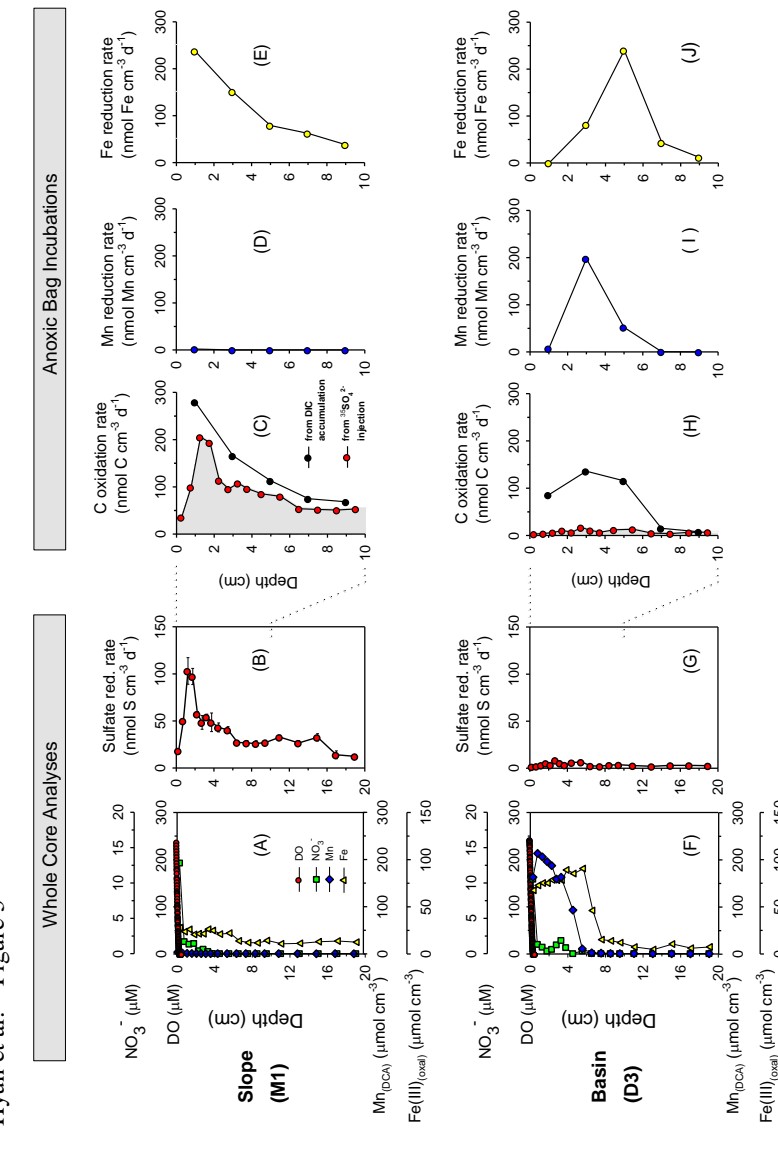

Fig. 5. Vertical distribution of terminal electron acceptors ($O_2$, $NO_3^-$, Mn and Fe) and rates of sulfate reduction measured from whole core analyses, and rates of anaerobic carbon oxidation (DIC production rates), Mn reduction and Fe reduction measured from anoxic bag incubations in Fig. 4. $C_{org}$ by sulfate reduction in panel C and H was calculated from the stoichiometry of 2:1 of $C_{org}$ oxidized to sulfate reduced.



Hyun et al. – Figure 6

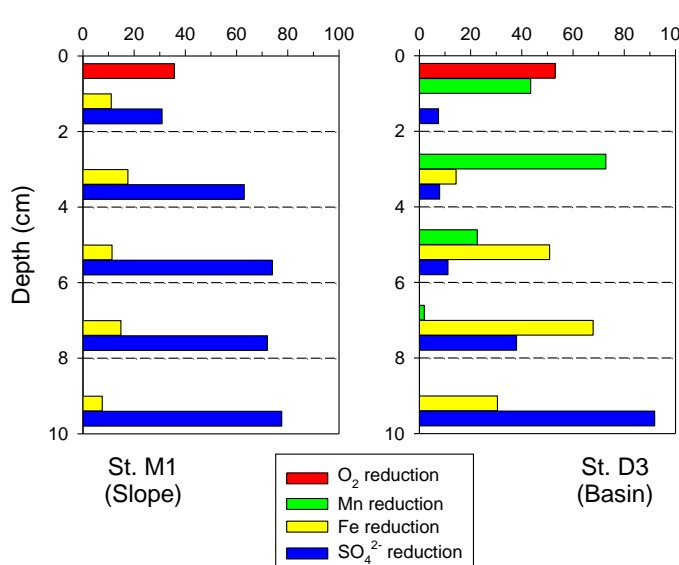

Fig. 6. Depth variations of partitioning of each carbon oxidation pathway in total carbon
oxidation at M1 and D3



Hyun et al - Figure 7

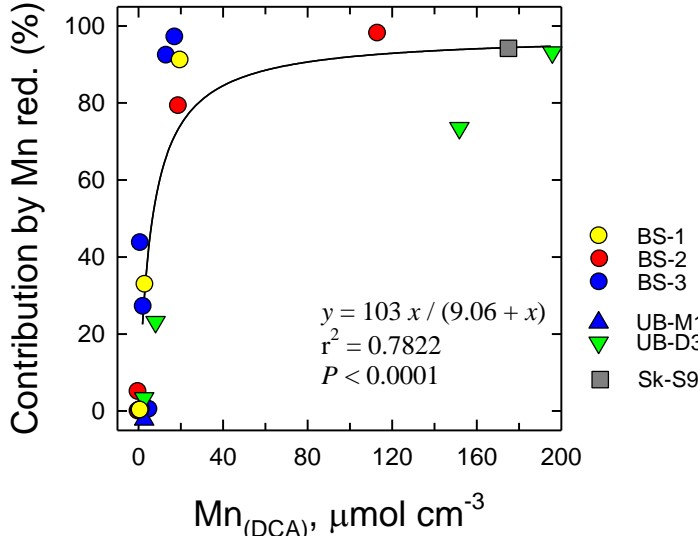



Fig. 7. The relative contribution of Mn reduction to total carbon oxidation as a function
of the concentration of Mn(DCA) at 3 different sites. BS, Black Sea (Thamdrup et al.
2000); UB, Ulleung Basin (This study); Sk, Skagerrak (Canfield et al. 1993b).
