# Peer review of "A manuscript submitted to discussion forum of BG as a BG discussion paper (bg-2016-222)"

_Biogeosciences, 2016_

## Referee Comment (RC1) · Anonymous Referee #1 · 10 Jul 2016

General comments

This paper proposes a method for quantifying the rates of organic C oxidation pathways in two deep continental margin sediment cores from the Ulleung Basin. It is one of the very few studies highlighting the role of Mn and Fe reduction as dominant organic C oxidation process in marine sediments. The study presents an excellent geochemical dataset on the sediment and sediment pore-water and on anoxic incubations of sediments. I really appreciated many aspects of the modeling such as the effort made to evaluate distinctly the O2 consumption for the organic matter oxidation and for the reoxidation of reduced species, as well as to assess the adsorption of Mn+2 on Mn oxi-hydroxides. The manuscript is well written and judiciously refers the reader to previous

works in the field. It assuredly deserves publication and will be of interest for a wide audience of aquatic geochemists.

Specific comments

Line 195: In addition to FeS and H2S, AVS also includes minute amounts of other metal sulfides. Isn't it?

Line 209: I presume that the modeling of the O2 micro-profiles with PROFILE was done assuming negligible bioirrigation and bioturbation but this is not specified. My deduction relies on the fact that a bioirrigation coefficient is not reported. However, these processes should not be insignificant since the authors state later in the MS that bioturbation realistically drives the Mn cycling in Ulleung sediments. I suggest clarifying this point.

Lines 246-249: It is not quite clear how the "abiotic Fe reduction coupled to H2S oxidation" was estimated with reaction (5)? Some clarification should be provided. Do the authors assume that AVS mainly equals FeS?

Line 606: The statement about the probable importance of bioturbation seems to be in contradiction with the well-defined utilization of the electron acceptors according to the order of decreasing energy yield for organic C oxidation that has been underscored in lines 412-417? Again, I suggest clarifying this point.

Minor technical corrections

Line 218: Provide the value of Do with a reference. . Line 221: Place in parentheses (see results section 3.2) after "...bimodal depth distribution".

Line 262: Madison et al. (2013) does not appear in the list of references.

Line 276: I suspect that the units (ml/g) are erroneous?

Lines 542-544: This sentence should be supported by references.

Please also note the supplement to this comment:
http://www.biogeosciences-discuss.net/bg-2016-222/bg-2016-222-RC1-
supplement.pdf

———————————————————

---

## Referee Comment (RC2) · S. Kasten (Referee) · 12 Jul 2016

Hyan et al. have investigated pathways and rates of organic carbon mineralization at two sites in the Ulleung Basin, East Sea. They have used a combination of direct and indirect approaches to determine rates of the different electron acceptor processes and compare these to the depth distribution of the different oxidants in the upper 10 cm of sediments of a slope site (station M1) and a site in the central Ulleung Basin (station D3). The authors demonstrate that metal oxide reduction represents the dominant pathway of organic carbon oxidation in the surface sediments of the deep central basin and suggest that Mn and Fe reduction may be more important in driving biogeochemical processes in similar marine settings than previously thought. Furthermore,

they point out that a significant increase in sea water (sea surface?) temperature is observed in the study area and that in this respect it is important to monitor the impact of this environmental change on biogeochemical cycles and element fluxes.

The authors have generated an excellent and unique data set and the manuscript is definitely suitable for and should be published in Biogeosciences after some minor revision (cf specific comments below). One of my major points is that I would suggest to better focus the objectives and conclusions. Do the authors really think that the phenomenon of Mn reduction/metal oxide reduction dominating organic carbon mineralization in marine sediments is a phenomenon of continental margin sediments in general? My understanding is rather that manganese reduction seems to be of particular importance in basin settings where reactive (partly freshly precipitated) Mn oxide phases are supplied in higher relative abundance due to geochemical focusing. Therefore, I recommend that the authors more clearly point out on which kind of depositional marine environment/s they are focusing on – continental margins, deep sea, deep basin settings?! (. . . and also highlight this in the title). Moreover, I think that a deep basin setting is not a typical continental margin depositional environment.

I also did not fully understand how this study may really contribute to monitoring the impact of environmental change (increase in surface water temperature) and how and why this future task is important in the framework of the present study. Do the authors assume that their investigation represents a kind of "baseline study" and will the detailed investigations performed here will be continued in the future? If yes, I would suggest to point this out more explicitly. On the other hand the authors have stated that the study area has (already) experienced the "fastest" (by the way: compared to what?) increase in sea water/sea surface temperatures. So, I assume that the study sites are already affected by this environmental change (because as you say – warming already started in 1982) and cannot really serve as a "pristine" (whatever that is) site or a location for a baseline study. So, if you mention the importance of assessing the impact of environmental changes on biogeochemical processes and cycles – please clearly

explain how this precisely relates to the current study – or just delete.

A further point is that references to more recent studies on organic carbon mineralization pathways and rates in deep-sea sediments are largely missing. Please, add the relevant more recent studies in the Introduction and Discussion chapters.

Specific comments

Title: the term "deep" suggests that you have worked in deep subsurface sediments – however, you have in fact studied surface sediments (i.e. the uppermost 10 cm). I would therefore suggest to rephrase the title to something like "Manganese and iron reduction dominate organic carbon oxidation in surface sediments of the deep Ulleung Basin"

L. 38: ... in "sediments of" the continental slope.

Ls. 50, 632 and 662: biogeochemical

L. 53: "fastest increase" compared to what exactly? Please explain.

Ls. 78/80: In this context I would also refer to and cite papers which demonstrate the close coupling of phosphate to the redox recycling of iron (e.g. Slomp etc.)

Ls. 84 ff: Numerous papers on rates and pathways of organic matter mineralization in carbon-starved deep-sea sediments have been published in recent years and should be cited here as well. Amongst others, these comprise work in the South Pacific Gyre in the framework of the IODP (e.g. D-Hondt et al. ) as well as in the Clarion–Clipperton fracture zone in the equatorial east Pacific Ocean (Mewes et al., 2014, Deep Sea Res., Part I, 91; Mewes et al., 2016, EPSL, 433; Mogollon et al., 2016, Geophys. Res. Lett., 43).

Ls. 87 ff.: I would suggest to rephrase and to add the recent reference by Bowles et al. (2014, Science) on global rates of sulfate reduction: ..., sulfate reduction "can" account for up to 50% of total carbon oxidation .... (.....; Bowles et al., 2014)

[Figure]

L. 95: . . . such as "the" Panama Basin

Ls. 101/102: "the" Japan Basin, "the" Yamato Basin and "the" Ulleung Basin . . .

L. 102: Rephrase to: . . ., the "surface waters" of the Ulleung Basin "are" characterized by higher . . .

L. 105: The enhanced biological production in the "euphotic zone of the Ulleung Basin" is responsible . . .

L. 115: "surface waters" instead of water column

L.118: . . . "at" the continental slope and rise . . ..

L. 132: delete "the" before two

L. 134: . . . continental "margin" sediments

L. 140: write: "in diameter". I would suggest that you also give the length of the sediment cores that you have investigated in the framework of this study.

L. 162: . . . in "a" N2-filled glove bag . . .

L. 164: "within" instead of in (4 weeks)

L. 174: Mn2+

Ls. 175 and 191: "inductively" coupled plasma-atomic emission spectrometry

L. 187: . . . "for the" (pore-water analysis) – instead of in

L. 192: What exactly do you mean with "free" Mn oxides? Please explain!

L. 194: What was the detection limit for Fe?

L. 230: "dissolved" instead of soluble

L. 283: Canfield et al. (1993a) were certainly not the first to give this stoichiometry.

[Figure]

L. 309 and throughout the manuscript: Please, always speak of "site" or "station" M1/D3.

L. 332 and throughout the manuscript: I would suggest to always speak of "contents" (instead of concentrations) when you refer to solid-phase values.

L. 339: "uptake" instead of utilization

L. 483: . . . of "the" respective electron acceptor —

L. 507: What exactly do you mean with "anoxia" here? Anoxic conditions in the surface sediments or anoxia in the water column (I assume you mean former).

L. 507 ff.: You describe here that there is a discrepancy between Mn2+ accumulation in the pore water and rates inferred from DIC accumulation and contribute this discrepancy to adsorption on Mn2+ to fresh Mn oxide surfaces. In this context we would like to bring to your attention our recently published paper on the coupling of manganese and nitrogen cycling in deep-sea sediments of the Clarion-Clipperton fracture zone in the equatorial Pacific Ocean (Mogollon et al., 2016, Geophys. Res. Lett., 43). In this paper we suggest that Mn2+ may act as a reducing agent for oxidized nitrogen species. Do you think that this process might potentially also play a role in the surface sediments of the Ulleung Basin? – in particular because the sample/depth interval where this discrepancy occurs is characterized by the presence of nitrate. Please discuss.

L. 531: "fall" instead of falls

Ls. 560 ff.: I do not understand this sentence at all. Please rephrase.

Ls. 564: In this context we would like to bring to your attention further studies on solid-phase manganese contents - Gingele and Kasten (1994, Mar. Geol., 121) and Mewes et al. (2014, Deep Sea Res., Part I, 91).

Ls. 578: With respect to the geochemical focusing of Mn in deep central parts of basin settings, I would also like to bring your attention to the work of Schaller et al. who have

worked on lake sediments and describe very similar accumulation mechanisms.

L. 611: Please delete "of" at the end of this line.

Ls. 617 and 623: . . . of "a" Chilean upwelling site

Ls. 619, 620 and 630: organic "carbon" flux/content

L. 637: . . . biogeochemical "cycles of" carbon . . .

L. 638: I still do not understand why the Ulleung Basin is a "biogeochemical hotspot"? Is it because organic matter mineralization is dominated by metal reduction? This is not clear at all and I would therefore suggest to better explain or to delete this.

I also do not like very much the statements you give with respect to the uniqueness of the study and the study site (eastern Asian marginal seas) in lines 547/548 and 646 ff. I think that you have performed a unique and excellent study and there is no need to justify this in terms of being the first study of this kind for this particular region.

L.650 ff: Is is possible that all Mn initially contained in the settling particles has been reductively mobilized from the deeper/buried sediments and has successively been concentrated in the uppermost sediments of the deep basin – leading to the observed high Mn contents? Please, discuss.

L. 656: Is Mn reduction really generally important in "deep-sea sediments"? Isn't it rather that Mn reduction/metal reduction is important in deep basin settings?

Ls. 664 ff: Fastest increase in sea water (I guess "surface water") temperature compared to what exactly? Please explain.

―――――――――――――――――――――

---

## Referee Comment (RC3) · Anonymous Referee #3 · 13 Jul 2016

The zonation of terminal electron accepting processes (TEAPs) in marine sediments and the importance of anaerobic TEAPs to the overall degradation of deposited organic carbon are concepts that all students of biogeochemistry learn very early in their careers. Nevertheless, as the authors point out, there are very few studies that attempt to quantitatively estimate the contributions of the various anaerobic dissimilatory processes to overall organic carbon decomposition. Employing a carefully calibrated set of biogeochemical rate measurements Hyun et al. tease apart the contribution of Mn, Fe and sulfate reduction to organic carbon degradation in the Ulleung Basin of the East Sea bordered by Korea, Russia and Japan. This builds on a research approach pioneered in the nineties by Thamdrup, Canfield and co-workers. Here, Hyun et al.

[Figure]

show that Mn reduction can be a powerful and important TEAP in during sedimentary organic matter degradation. While there are a number of assumptions built into the experiments used to estimate dissimilatoryFe and Mn reduction rates, this is still state of the art approach in sedimentary biogeochemistry. Moreover, from such experiments, they conclude that Mn reduction may be underestimated as a process in marine sediments as the techniques for studying Fe and Mn reduction require thorough and careful experimentation, as demonstrated in this study. Hyun et al.'s study is a further, solid step in the right direction and stands in stark contrast to recent efforts that only examine pore water concentration profiles (e.g. Bowles et al., Science 2014), which underestimate or miss the contribution of Fe, Mn and sulfate reduction towards organic carbon decomposition in near-surface sediments.

I especially appreciated the attempt to link the relative contribution to total carbon oxidation as a function of the Mn content of the sediment. In general, this is a solid contribution to marine sediment biogeochemistry and I look forward to using these results as an excellent example of anaerobic TEAP processes in classes for the next generation of biogeochemistry students.

Specific comments:

Line 91: The authors might want to consider estimates from D'Hondt et al. Science 2004, where areal rates of Mn and Fe reduction in the Peru Basin have been estimated. Although derived from deep sub-surface pore water profiles, it would be interesting to see how the Peru Basin site maps onto Figure 7.

Line 169: Pore water analysis....can approximate detection limits be provided?

Line 249 and following : It would be helpful to have a formula that includes the effect of SR the estimate of dissimilatory Fe (III) reduction (Equation 4) that includes the stoichiometry from Equation 5.

Line 376...."that" does not take a comma.

[Figure]

Line 388 and 414: Are "evidenced" and "zonated" proper verbs?

Line 440 "Consequently" not necessary

Line 489 Replace "this" with "these"

Paragraph at line 498: I find this argument to be a bit of a stretch. Basically the authors are saying because Canfield et al. had a negative result with a complexed Fe experiment that they also have no effect of ferrous iron on MnO4 reduction. This is a bit weak. The argument following at line 505 is stronger.

Paragraph starting at Line 548: I found this part very interesting. What happens when one moves out into total oxic sedimentary environments where oxygen penetration is deep and there are plenty of Mn oxide crusts (deep ocean sediment)? Should perhaps the Figure 7 refer to relative contribution of Mn reduction to "anaerobic" carbon oxidation?

Line 594: The authors might point out that they are probably underestimating total oxygen uptake. Bioirrigation might very likely play a major role in these sediments.

---

## Author Comment (AC1) · 7 Oct 2016

Response on the interactive comment on "Manganese and iron reduction dominate organic carbon oxidation in deep continental margin sediments of the Ulleung Basin, East Sea" by Jung-Ho Hyun et al.

Anonymous Referee #1

General comments

This paper proposes a method for quantifying the rates of organic C oxidation pathways in two deep continental margin sediment cores from the Ulleung Basin. It is one of the very few studies highlighting the role of Mn and Fe reduction as dominant organic C

oxidation process in marine sediments. The study presents an excellent geochemical dataset on the sediment and sediment pore-water and on anoxic incubations of sediments. I really appreciated many aspects of the modeling such as the effort made to evaluate distinctly the O2 consumption for the organic matter oxidation and for the reoxidation of reduced species, as well as to assess the adsorption of Mn+2 on Mn oxihydroxides. The manuscript is well written and judiciously refers the reader to previous works in the field. It assuredly deserves publication and will be of interest for a wide audience of aquatic geochemists. (General response): 1. Thank you very much for this highly positive comments on the manuscript. We will try to incorporate your comments as much as possible in the revisied version of the manuscript. 2. In this revised manuscript, we have modified the title of the manuscript to: "Manganese and iron reduction dominate organic carbon oxidation in surface sediments of the deep Ulleung Basin, East Sea" to clarify that the dominance of Mn and Fe reduction occurs in the surface sediments of the deep basin.

Specific comments

1) Line 195: In addition to FeS and H2S, AVS also includes minute amounts of other metal sulfides. Isn't it? (Response) Yes. AVS includes several dissolved and solid-phase constituents such as H2S, FeS, iron sulfide nanoparticles and other metal sulfides (Rickard and Morse, 2005. Mar Chem 97: 141-197; Luther, 2005. Mar Chem 97: 198-205). Since FeS nanoparticles and FeSaq molecular clusters generally contribute only a small fraction of total AVS (Luther, 2005) and the content of other metals such as Zn and Cu is normally much lower than that of Fe, we assume that this is also the case here.

(Correction): We will change the sentence to "For the determination of total reduced sulfur (TRS) that includes acid volatile sulfide (AVS = FeS + H2S and small amounts of other metal sulfides, see Rickard and Morse, 2005; Luther, 2005) and chromium-reducible sulfur (CRS = S0 + FeS2), $\sim\sim$" in line 203 – 205 in the revised manuscript.

2) Line 209: I presume that the modeling of the O2 micro-profiles with PROFILE was done assuming negligible bioirrigation and bioturbation but this is not specified. My deduction relies on the fact that a bioirrigation coefficient is not reported. However, these processes should not be insignificant since the authors state later in the MS that bioturbation realistically drives the Mn cycling in Ulleung sediments. I suggest clarifying this point.

Response: This is correct. Bioirrigation can contribute substantially to total oxygen uptake in some sediments, while sediment reworking through bioturbation is mainly of importance for the transport of solids. Thus, bioturbation coefficients are typically at least an order of magnitude lower than the molecular diffusion coefficient for oxygen (the biodiffusion coefficient estimated here (9.5 cm2 y-1) is ~50 times lower, but may yet be very important in the cycling of solids (e.g., Boudreau 1994, GCA 58:1243; see also response to comment #4). Bioirrigation has not been investigated in these sediments. We have discussed the potential underestimation of oxygen uptake and the consequences for the conclusions based on the large dataset reviewed by Glud (2008; Mar Biol Res 4: 243-289).

(Correction): 1. Finally, we have corrected our quantative estimation on the partitioning of Corg oxidation pathways at each station in line 476 – 488: "Additionally, our partitioning of carbon oxidation pathways could be biased towards the anaerobic electron acceptors due to the use of the diffusive oxygen uptake (DOU) rather than total oxygen uptake (TOU), which will exceed DOU if bioirrigation is active (Glud 2008). Bioirrigation was not determined at our sites, but the pore water profiles show no indication of strong irrigation (Fig. 2). An average DOU/TOU ratio of ~0.6 has been reported for sediments at 1.5 – 2.5 km depth (Glud 2008). Using this ratio, and assuming that TOU is partitioned similarly as DOU between aerobic carbon oxidation and reoxidation, aerobic carbon oxidation would account for 25%, while Fe and sulfate reduction would account for 11% and 46% of of carbon oxidation, respectively. This, the potential bias from using DOU is not expected to affect the ranking of electron acceptors by quantitative importance ($SO_4^{2-}$ > $O_2$ > Fe(III)), and, as discussed further below, the partitioning of Corg oxidation at M1 falls within the range previously reported for continental margin sediments."

2. We also discussed the significance of aerobic respiration at basin site (D3) in line 505 – 507 in revised manuscript: "Correction for a potential underestimation of TOU, as discussed for M1, would reduce the contributions of Mn and Fe reduction slightly to 41% and 18%, respectively."

3) Lines 246-249: It is not quite clear how the "abiotic Fe reduction coupled to H2S oxidation" was estimated with reaction (5)? Some clarification should be provided. Do the authors assume that AVS mainly equals FeS? (Response): The procedure used was adopted from Gribsholt et al. (2003). The rationale is that in the presence of reactive Fe(III), H2S produced from sulfate reduction reacts with Fe(III) and the $Fe^{2+}$ produced from this reaction precipitates as FeS. The stoichiometry of abiotic reduction of 2Fe coupled to reaction of 3H2S in marine sediments is presented in several studies. For example, Melton et al. (2014, Nature Reviews. Microbiol.12: 797 – 808) stated that "At neutral pH, hydrogen sulphide (H2S) can abiotically reduce Fe(III) oxyhydroxides: $2FeOOH + 3H_2S \rightarrow 2FeS + S^0 + 4H_2O$. H2S reactions with Fe are especially important in marine environments, where high sulphate concentrations and microbial S reduction lead to pronounced H2S production". So, the H2S in the equation 5 represents the H2S produced from the sulfate reduction that we directly measured using 35S injection incubation exeriment.

(Correction): We have changed the equation (5) to "$2FeOOH + 3H_2S$(produced by SR) $= 2FeS + So + 4H_2O$, and add one more review paper as a reference (Melton et al., 2014) in line 256 – 259 in revised manuscript.

4) Line 606: The statement about the probable importance of bioturbation seems to be in contradiction with the well-defined utilization of the electron acceptors according to the order of decreasing energy yield for organic C oxidation that has been underscored in lines 412-417? Again, I suggest clarifying this point. (Response): With bioturbation being a diffusive process, as assumed here and in other studies, we see no contradiction between the presence of bioturbation and the relatively distinct redox zonation. The estimated biodiffusion coefficient of (Db) of 9.5 cm2 yr-1 at Site D3 corresponds to ∼2% of the molecular diffusion coefficient of oxygen (388 cm2 yr-1 ). Judging from the absence of major fauna in the UB sediments, the mixing is brought about by small organisms with each individual affecting only a small area relative to the size of our cores, and the Db averaging many of these small but frequent events. Similarly, e.g., Hyacinthe et al (2001) found that well defined profiles can be observed in both sediments with low and high bioactivity in the Bay of Biscay. Therefore, we were able to see the well defined zonation of electron acceptors (Fig. 5F) in the UB where bioturbation is relatively weak.

Minor technical corrections

1) Line 218: Provide the value of Do with a reference. (Response): We will provide the value of Do with reference as follows "where Do (1.07 ïĆť 10-9 m2 s-1 at M1 and 1.03 ïĆť 10-9 m2 s-1 at D3) is the temperature-corrected molecular diffusion coefficient estimated from Schulz (2006)" in line 227 – 228 in the revised manuscript.

2) Line 221: Place in parentheses (see results section 3.2) after ". . .bimodal depth distribution". (Response): We wil place it as follows "∼∼ bimodal depth distribution (see results section 3.2) ∼∼". in line 231 - 232 in the revised manuscript

3) Line 262: Madison et al. (2013) does not appear in the list of references. (Response): Thank you. We will add the reference in the reference list in revised manuscript.

4) Line 276: I suspect that the units (ml/g) are erroneous? (Response): It is presented in Thamdrup et al (2000).

5) Lines 542-544: This sentence should be supported by references. (Response): Yes.

We have added references (i.e., Canfield et al., 1993b; Thamdrup et al., 2000) in the sentence (line 569-570).

Please also note the supplement to this comment: http://www.biogeosciences-discuss.net/bg-2016-222/bg-2016-222-RC1- supplement.pdf Interactive comment on Biogeosciences Discuss., doi:10.5194/bg-2016-222, 2016.

Please also note the supplement to this comment:
http://www.biogeosciences-discuss.net/bg-2016-222/bg-2016-222-AC1-
supplement.pdf

[Figure]

**Supplement:**

[revised manuscript text omitted]
 Seais often called as "a miniature ocean" because of the independent thermohaline convection system that is driven by the high density surface water sinking (Kim et al., 2001) in a manner similar to that of the Great Ocean Conveyor Belt (Broecker, 1991).

The turnover time (ca. 100 – 300 years) of the thermohaline circulation is shorter than that of the global conveyor belt of 1000 – 2000 years (Broecker and Peng, 1982). Because of the shorter time-scale, together with the relatively small volume, the East Sea is expected to be much more sensitive to global environmental changes (such as global warming) compared with the open oceans. In this regard, the East Sea has been considered as a natural laboratory that provides a useful field for large-scale oceanographic experiments to predict the response of oceans associated with long-term climatic/oceanographic changes (Kim et al., 2001). Over the last two decades (1982 – 2006), a rapid increase of sea surface temperature (SST) of

1.09 ℃ has been recorded in the East Sea, which is the fourth highest among the 18 large marine ecosystems in the world ocean (Belkin, 2009). Increased SST reduces the soubility of

$O_2$ in the surface mixed layer and enhances stratification, which ultimately affects biological production in the water column and suppresses transport of $O_2$-rich surface water into the deep bottom. Indeed, recent oeanographic 
[revised manuscript text omitted]

---

## Author Comment (AC2) · 7 Oct 2016

Response on the interactive comment on "Manganese and iron reduction dominate organic carbon oxidation in deep continental margin sediments of the Ulleung Basin, East Sea" by Jung-Ho Hyun et al.

S. Kasten (Referee) Sabine.Kasten@awi.de General comments

1) Hyun et al. have investigated pathways and rates of organic carbon mineralization at two sites in the Ulleung Basin, East Sea. They have used a combination of direct and indirect approaches to determine rates of the different electron acceptor processes and compare these to the depth distribution of the different oxidants in the upper 10 cm of sediments of a slope site (station M1) and a site in the central Ulleung Basin (station D3). The authors demonstrate that metal oxide reduction represents the dominant pathway of organic carbon oxidation in the surface sediments of the deep central basin and suggest that Mn and Fe reduction may be more important in driving biogeochemical processes in similar marine settings than previously thought. Furthermore, they point out that a significant increase in sea water (sea surface?) temperature is observed in the study area and that in this respect it is important to monitor the impact of this environmental change on biogeochemical cycles and element fluxes. The authors have generated an excellent and unique data set and the manuscript is definitely suitable for and should be published in Biogeosciences after some minor revision (cf specific comments below). (General response): 1. Thank you very much for your positive and critical comments that will improve the quality of this manuscript. I together with my co-autors will try to incorporate your comments as much as we can. 2. In this revised manuscript, as you suggested, we have modified the title of the manuscript to: "Manganese and iron reduction dominate organic carbon oxidation in surface sediments of the deep Ulleung Basin, East Sea" to clarify that the dominance of Mn and Fe reduction occurs in the surface sediments of the deep basin.

2) One of my major points is that I would suggest to better focus the objectives and conclusions. Do the authors really think that the phenomenon of Mn reduction/metal oxide reduction dominating organic carbon mineralization in marine sediments is a phenomenon of continental margin sediments in general? My understanding is rather that manganese reduction seems to be of particular importance in basin settings where reactive (partly freshly precipitated) Mn oxide phases are supplied in higher relative abundance due to geochemical focusing. Therefore, I recommend that the authors more clearly point out on which kind of depositional marine environment/s they are focusing on – continental margins, deep sea, deep basin settings?! (: : : and also highlight this in the title). Moreover, I think that a deep basin setting is not a typical continental margin depositional environment.

(Response): Thank you for this thoughtful comment. As you pointed out, if we consider the depth > 2000 m, the UB is a deep basin sediment, and the dominance of Mn reduction is of importance at the deep basin site. However, the East Sea including the Ulleung Basin is regarded a marginal sea since it is isolated from the open ocean (northwest Pacific), and is surrounded by Koran peninsula and Japan islands (Kang et al. 2010; Liu et al., 2010). On the other hand, as you pointed out, the dominance of Mn reduction is of importance in the basin site (D3), and is not significant for Corg oxidation processes at all in the slope site (M1). In this regard, I see that the title you suggested fits perfectly to resolve the two contradicting aspects.

Concerning the generalization of the results to other environments, the discussion (end of Section 4.2) is based mainly on the relationship of Mn reduction to sediment Mn oxide content (Fig. 7), which includes both continental margin sites and deep basins. Also the citations included as examples of locations with sufficiently high Mn oxide content for Mn reduction to make a substantial contribution to carbon oxidation include both continental margin and deep basin sites. While some of these sites are not clearly located in basins, we do agree that bottom topography is likely to play an important role in allowing geochemical focusing.

(Correcton): 1. We will change the title of the manuscript as you recommended in revised manuscript (please see the response on your specific comment #1).

2. We will add more description on the UB in the Study Site section in line 129 – 134 in the revised manuscript as follows "The East Sea is a marginal sea surrounded by the east Asian continent and Japanease Islands (Fig.1, Kang et al., 2010). The UB located in the southwestern part of the East Sea is a bowl-shaped deep basin (2000 – 3000 m depth) (Fig. 1) delimited by continental slopes of Korean Peninsula and the southwestern Japanese Archipelago on the west and south, respectively, and by the Korea Plateau and the Oki Bank on the north and east, respectively (Chough et al., 2000)".

3. We will also change the Discussion to clarify that Mn reduction is mainly expected to be important in basin settings in line 590-596 in the revised manuscript as follows: "Manganese enrichments of this magnitude have been reported for several locations on the continental margins and in deep basins (Murray et al., 1984; Gingele and Kasten, 1994; Gobeil et al., 1997; Haese et al., 2000; Mouret et al., 2009; Magen et al., 2011; Macdonald and Gobeil, 2012; Mewes et al., 2014) in addition to the relatively few places where dissimilatory Mn reduction was already indicated to be of importance, as discussed above. Thus, the process may be of more widespread significance, particularly in deep basin settings such as UB that allow geochemical focusing of manganese". (Please see also the response to comment #43.)

3) I also did not fully understand how this study may really contribute to monitoring the impact of environmental change (increase in surface water temperature) and how and why this future task is important in the framework of the present study. Do the authors assume that their investigation represents a kind of "baseline study" and will the detailed investigations performed here will be continued in the future? If yes, I would suggest to point this out more explicitly. On the other hand the authors have stated that the study area has (already) experienced the "fastest" (by the way: compared to what?) increase in sea water/sea surface temperatures. So, I assume that the study sites are already affected by this environmental change (because as you say – warming already started in 1982) and cannot really serve as a "pristine" (whatever that is) site or a location for a baseline study. So, if you mention the importance of assessing the impact of environmental changes on biogeochemical processes and cycles – please clearly explain how this precisely relates to the current study – or just delete.

(Response): We are sorry for this confusion. I have skipped too much in explaining the background of this last section, and I am glad to have a chance to refine the significance of monitoring the vaiations of the rate and pathways of Corg oxidation (i.e., monitoring relative significance of each TEAP) with the decreasing O2 concentrations and temperature increase in the bottom layer of the UB (or East Sea) in predicting and understanding future biogeochemical processes in the East Sea. First, both the points raised here (i.e., baseline study and long-term monitoring) are correct. We see the measurements as a contribution to defining the current state as a baseline for monitoring potential future change, but we also acknowledge that some changes may have already taken place for last several decades.

(Correction): We will explain these aspects better in Discussion (line 663 – 688 in SECTION 4.4) in the revised manuscript as follows: "The East Sea is often called as "a miniature ocean" because of the independent thermohaline convection system that is driven by the high density surface water sinking (Kim et al., 2001) in a manner similar to that of the Great Ocean Conveyor Belt (Broecker, 1991). The turnover time (ca. 100 – 300 years) of the thermohaline circulation is shorter than that of the global conveyor belt of 1000 – 2000 years (Broecker and Peng, 1982). Because of the shorter time-scale, together with the relatively small volume, the East Sea is expected to be much more sensitive to global environmental changes (such as global warming) compared with the open oceans. In this regard, the East Sea has been considered as a natural laboratory that provides a useful field for large-scale oceanographic experiments to predict the response of oceans associated with long-term climatic/oceanographic changes (Kim et al., 2001). Over the last two decades (1982 – 2006), a rapid increase of sea surface temperature (SST) of 1.09 °C has been recorded in the East Sea, which is the fourth highest among the 18 large marine ecosystems in the world ocean (Belkin, 2009). Increased SST reduces the soubility of $O_2$ in the surface mixed layer and enhances stratification, which ultimately affects biological production in the water column and suppresses transport of $O_2$-rich surface water into the deep bottom. Indeed, recent oeanographic observations revealed that the gradual deoxygenation and warming of the bottom water of the East Sea over the last 30 years have resulted in an âĹij10% decrease in dissolved oxygen and âĹij0.04 °C increase in potential temperature, which suggested a weakening of the deep convection system (Kim et al., 2001; Gamo et al., 2011). Benthic metabolism and respiratory $C_{org}$ oxidation coupled to various TEAP in the sediments are largely controlled by the combination of $O_2$ content, temperature and biological production overlying water column (Canfield et al., 2005). It is thus important to monitor any changes in the rates and partitioning of Corg oxidation to better understand and predict the variations of biogeochemical cycles of carbon, nutrients and metals potentially associated with long-term climatic changes in the UB, the biogeochemical hotspot of the East Sea.."

4) A further point is that references to more recent studies on organic carbon mineralization pathways and rates in deep-sea sediments are largely missing. Please, add the relevant more recent studies in the Introduction and Discussion chapters. (Response): Thank you for this comment, and providing those references. We will the references in Introduction and Discussion. Please see the response on your specific comment # 6, 7, 31 and 35.

Specific comments

1) Title: the term "deep" suggests that you have worked in deep subsurface sediments – however, you have in fact studied surface sediments (i.e. the uppermost 10 cm). I would therefore suggest to rephrase the title to something like "Manganese and iron reduction dominate organic carbon oxidation in surface sediments of the deep Ulleung Basin" (Response): Thanks for the comment. We will change the title in revised manuscript as you suggested.

2) L. 38: : : : in "sediments of" the continental slope. (Response): Yes, we have changed this in line 38.

3) Ls. 50, 632 and 662: biogeochemical (Response): Thank you for the correction. We have corrected the mistyping in line 50, 661, and 710.

4) L. 53: "fastest increase" compared to what exactly? Please explain. (Response): We have changed it to " $\sim$ where the gradual deoxygenation and warming of the bottom water have resulted in an âĹij10% decrease in dissolved oxygen and âĹij0.04 oC increase in potential temperature for the past three decades." in line 53 – 55, and more explanation is presented in line 663 – 688 in discussion section of the revised manuscript (Please see the response and correction the we made for your general comment #3).

5) Ls. 78/80: In this context I would also refer to and cite papers which demonstrate the close coupling of phosphate to the redox recycling of iron (e.g. Slomp etc.) (Response): Thank you very much for this appropriate comments. We will add two more references in line 76 – 77 as you suggested.

Hensen et al., 2006 Benthic cycling of oxygen, nitrogen, and phosphorus (Marine geochemistry Ch.6). Slomp et al., 2013 Coupled dynamics of iron and phosphorus in sediments of an oligotrophic coastal basin and the impact of anaerobic oxidation of methane

6) Ls. 84 ff: Numerous papers on rates and pathways of organic matter mineralization in carbon-starved deep-sea sediments have been published in recent years and should be cited here as well. Amongst others, these comprise work in the South Pacific Gyre in the framework of the IODP (e.g. D-Hondt et al. ) as well as in the Clarion–Clipperton fracture zone in the equatorial east Pacific Ocean (Mewes et al., 2014, Deep Sea Res., Part I, 91; Mewes et al., 2016, EPSL, 433; Mogollon et al., 2016, Geophys. Res. Lett., 43). (Response) We will cite those references (D'Hondt et al., 2015; Mewes et al. 214, 2016) in the revised manuscript, and revised the sentence to "In general, aerobic metabolism dominates the organic matter mineralization in deep-sea sediments that are characterized by low organic matter content (Jahnke et al., 1982; Glud, 2008), especially in organic carbon-starved deep-sea sediments with low sedimentation rates (Mewes et al., 2014, 2016; D'Hondt et al. 2015; Mogollón et al., 2016)." in line 81 – 85.

7) Ls. 87 ff.: I would suggest to rephrase and to add the recent reference by Bowles et al. (2014, Science) on global rates of sulfate reduction: : : :, sulfate reduction "can" account for up to 50% of total carbon oxidation : : :. (: : :..; Bowles et al., 2014) (Response): Yes. We have added the reference in line 88.

8) L. 95: : : : such as "the" Panama Basin (Response): Thanks. We have corrected in line 95.

9) Ls. 101/102: "the" Japan Basin, "the" Yamato Basin and "the" Ulleung Basin : : : (Response): Thanks. We have corrected in line 101 – 102.

10) L. 102: Rephrase to: : : :, the "surface waters" of the Ulleung Basin "are" characterized by higher : : : (Response): I have rephrased this in line 102 – 103.

11) L. 105: The enhanced biological production in the "euphotic zone of the Ulleung Basin" is responsible : : : (Response): We have rephrased it as you suggested in line 105 – 106.

12) L. 115: "surface waters" instead of water column (Response): Correction has been made in line 116 as you suggested.

13) L.118: : : : "at" the continental slope and rise : : :. (Response): Correction has been made in line 119

14) L. 132: delete "the" before two (Response): Yes. We did it in line 138.

15) L. 134: : : : continental "margin" sediments (Response): Yes. We have added the word in line 141.

16) L. 140: write: "in diameter". I would suggest that you also give the length of the sediment cores that you have investigated in the framework of this study. (Response): Yes! I did it according to your suggestion in line 146 – 147.

17) L. 162: : : : in "a" N2-filled glove bag : : : (Response): Correction has been done in line 168.

18) L. 164: "within" instead of in (4 weeks) (Response): Correction has been made in line 170.

19) L. 174: Mn2+ (Response): We changed it to Mn2+ in line 180

20) Ls. 175 and 191: "inductively" coupled plasma-atomic emission spectrometry (Response): Thanks. Corrections have been made in line 181 and 199.

21) L. 187: : : : "for the" (pore-water analysis) – instead of in (Response): Thanks. We hve made a correction in line 195.

22) L. 192: What exactly do you mean with "free" Mn oxides? Please explain! (Response): Dithionite is used for extraction of all forms of iron and manganese oxides that are not protected (i.e., "free") by an insoluble barrier such as a silica concretion surrounding oxide particles. Free oxides is a term used traditionally in studies of iron and manganese partitioning in soils and, later, sediments (see, e.g., Canfield 1989. GCA 53:619; Golden et al. 1994. Clay and Clay Minerals. 42: 53-62).

23) L. 194: What was the detection limit for Fe? (Response): We have added the detection limit of the Fe2+ (1 $\mu$M) in line 202.

24) L. 230: "dissolved" instead of soluble (Response): We changed it to "dissolved" in line 240.

25) L. 283: Canfield et al. (1993a) were certainly not the first to give this stoichiometry. (Response): Honestly, I do not know the original paper that presented the equaitons first. Basically, those equations assuming an oxidation state of zero for the organic carbon can be found easily in several books and papers (e.g., Froelich et al. 1979). I chose the Canfield et al. (1993a), and hope that is fine with you.

26) L. 309 and throughout the manuscript: Please, always speak of "site" or "station" M1/D3. (Response): To make the manuscript more concise, We renamed the "Station M1" to "M1" and "Station D3" to "D3" in the Study area section in revised manuscript (line 139-140). Otherwise, there are too many places to express St. D3 or St. M1, which seems to be too redundant. I hope it is okay for you.

27) L. 332 and throughout the manuscript: I would suggest to always speak of "contents" (instead of concentrations) when you refer to solid-phase values. (Response):

Thank you for the comment. We will change te solid-phase values to "contents" in revised manuscript.

28) L. 339: "uptake" instead of utilization (Response): Yes. We will change it to "uptake" in line 348

29) L. 483: : : : of "the" respective electron acceptor — (Response): Yes. We have added "the" in line 502.

30) L. 507: What exactly do you mean with "anoxia" here? Anoxic conditions in the surface sediments or anoxia in the water column (I assume you mean former). (Response): You are right. I changed the phrase to "Despite the anoxic condition and nitrate depletion during the sediment incubation," ∼∼. in line 528 in the revised manuscript.

31) L. 507 ff.: You describe here that there is a discrepancy between Mn2+ accumulation in the pore water and rates inferred from DIC accumulation and contribute this discrepancy to adsorption on Mn2+ to fresh Mn oxide surfaces. In this context we would like to bring to your attention our recently published paper on the coupling of manganese and nitrogen cycling in deep-sea sediments of the Clarion-Clipperton fracture zone in the equatorial Pacific Ocean (Mogollon et al., 2016, Geophys. Res. Lett., 43). In this paper we suggest that Mn2+ may act as a reducing agent for oxidized nitrogen species. Do you think that this process might potentially also play a role in the surface sediments of the Ulleung Basin? – in particular because the sample/depth interval where this discrepancy occurs is characterized by the presence of nitrate. Please discuss. (Response): Thank you for the suggestion and the references associated with the Mn2+ oxidation coupled to NOx reduction. The process would not explain the absence of Mn2+ accumulation in the anoxic incubations, because nitrate was rapidly depleted, but it could play a role in situ. We will add to the discussion as follows "Low Mn2+ together with the rapid decrease of nitrate at 0-2 cm depth at D3 (Fig. 2F, 2G) also suggested that dissolved reduced manganese might act as a reduc-

ing agent for nitrate as it was suggested by Aller et al. (1998) in the Panama Basin and Mogollón et al. (2016) in the deep-sea sediment of the Clarion-Clierton fracture zone in the northeast equatirial Pacific." in line 533 – 537 in the revised manuscript.

32) L. 531: "fall" instead of falls (Response): Thanks. We did it in line 557.

33) Ls. 560 ff.: I do not understand this sentence at all. Please rephrase. (Response): The sentence says that there is a good relationship between MnO2 content and the contribution of Mn reduction, and that the contribution of Mn reduction is important even at low MnO2 content. And additional discussion is presented after the sentence. As it seems to be confusing, as you pointed out, we have changed it to "The plot indicates a close correlation between Mn oxide content and the importance of Mn reduction. Curve-fitting yields a concentration of MnO2 at 50 % of contribution of manganese reduction to total Corg oxidation (Ks) of 8.6 $\mu$mol cm-3 similar to the approx. 10 $\mu$mol cm-3 suggested before (Thamdrup et al., 2000). This indicates that Mn reduction can be a dominant Corg oxidation process even at low concentrations of Mn oxides compared to those found at UB" in line 584 – 589 of the revised manuscript.

34) Ls. 564: In this context we would like to bring to your attention further studies on solidphase manganese contents - Gingele and Kasten (1994, Mar. Geol., 121) and Mewes et al. (2014, Deep Sea Res., Part I, 91). (Response): Thanks for providing the references on the solid phase Mn. We have included them in line 591 – 593.

35) Ls. 578: With respect to the geochemical focusing of Mn in deep central parts of basin settings, I would also like to bring your attention to the work of Schaller et al. who have worked on lake sediments and describe very similar accumulation mechanisms. (Response): Yes. I have added the "Schaller and Wehrli (1997)" that fits well in the sentence in line 609 in the revised manuscript.

36) L. 611: Please delete "of" at the end of this line. (Respnse): Yes. We have deleted "of" in line 640.

37) Ls. 617 and 623: : : : of "a" Chilean upwelling site (Response): Yes. We have added the "a" in line 646 and 652.

38) Ls. 619, 620 and 630: organic "carbon" flux/content (Response): Yes. I have added "carbon" in line 649 and 659 in the revised manuscript

39) L. 637: : : : biogeochemical "cycles of" carbon : : : (Response): Yes. I have changed it in line 686 in the revised manuscript.

40) L. 638: I still do not understand why the Ulleung Basin is a "biogeochemical hotspot"? Is it because organic matter mineralization is dominated by metal reduction? This is not clear at all and I would therefore suggest to better explain or to delete this. (Response): It is not because organic carbon mineralization is dominated by metal reduciton. The reason that we stated the UB as a biogeochemical hot spot is that the overall organic carbon oxidation in the UB is higher than those measured in major upwelling system such as Benguela upwelling system and is even comparable to those reported at the continental slope of the Chilean upwelling system at a similar depth range of 1000 – 2500 m. Please see the line 640 – 662 in the revised manuscript.

41) I also do not like very much the statements you give with respect to the uniqueness of the study and the study site (eastern Asian marginal seas) in lines 547/548 and 646 ff. I think that you have performed a unique and excellent study and there is no need to justify this in terms of being the first study of this kind for this particular region. (Response): As you wished, we will delete the sentence in line 646 in original version (i.e., For the first time in the Asian marginal seas, and in one of only few experimental studies of the partitioning of Corg oxidation pathways in deep-sea sediments in general, we∼). We feel that the statement in line 570-573 in the revised manuscript makes the sentence stronger and more determinant and fluent. I hope this is fine with you.

42) L.650 ff: Is is possible that all Mn initially contained in the settling particles has been reductively mobilized from the deeper/buried sediments and has successively been concentrated in the uppermost sediments of the deep basin – leading to the observed high Mn contents? Please, discuss. (Response): Yes, our interpretation is that essentially all Mn is stripped from the settling particles during burial, and the surface enrichment further requires that the Mn oxides that form from reoxidation of Mn2+ in the surface sediment are also reduced and recycled over and over. Nonetheless, at steady state, the net sedimentation and burial fluxes should be the same. We added some text to clarify this in line 612 – 619 in the revised manuscript as follows: "Adopting the sediment accumulation rate of 0.07 cm y-1 in the UB determined at a station 50 km from D3 (Cha et al., 2005), the average Mn(DCA) concentration of 1.1 $\mu$mol cm-3 at 10 – 20 cm depth (Fig. 2G) corresponds to a flux for permanent burial of 0.002 mmol m-2 d-1 or just 0.03 % of the Mn reduction rate (Table 3), i.e., an Mn atom is recycled 3800 times before it finally gets buried – first by stripping from the particles that settle to the seafloor and subsequently, over and over, by reductive dissolution of the Mn oxides that from by reoxidation in the oxic surface layer (or, potentially, in the nitrate zone; Aller et al. 1998, Mogollón et al. 2016)".

43) L. 656: Is Mn reduction really generally important in "deep-sea sediments"? Isn't it rather that Mn reduction/metal reduction is important in deep basin settings? (Response): Yes – we have changed this to deep basin sediments. As mentioned in the response to comment #2, Mn enrichments also appear widespread in continental marign sediments. However, the present study mainly focuses on the deep basin setting, and we have therefore chosen to emhasize this type of environment in the conclusion. (Correction): We will change the sentence to " $\sim$ thereby that the process might be more important in continental margin and deep basin sediments than previously thought" in line 702 – 704 in the revised manuscript.

44) Ls. 664 ff: Fastest increase in sea water (I guess "surface water") temperature compared to what exactly? Please explain. (Response): We will replace this sentence with "Over the last 30 years, the gradual deoxygenation and warming of the bottom water of the East Sea have resulted in an âĹij10% decrease in dissolved oxygen and âĹij0.04âŲęC increase in potential temperature" in line 711 – 713 in the revised manuscript. (For more explanation, please see the response and correction made to your general coment #3.)

Please also note the supplement to this comment:
http://www.biogeosciences-discuss.net/bg-2016-222/bg-2016-222-AC2-supplement.pdf

**Supplement:**

[revised manuscript text omitted]
 Seais often called as "a miniature ocean" because of the independent thermohaline convection system that is driven by the high density surface water sinking (Kim et al., 2001) in a manner similar to that of the Great Ocean Conveyor Belt (Broecker, 1991).

The turnover time (ca. 100 – 300 years) of the thermohaline circulation is shorter than that of the global conveyor belt of 1000 – 2000 years (Broecker and Peng, 1982). Because of the shorter time-scale, together with the relatively small volume, the East Sea is expected to be much more sensitive to global environmental changes (such as global warming) compared with the open oceans. In this regard, the East Sea has been considered as a natural laboratory that provides a useful field for large-scale oceanographic experiments to predict the response of oceans associated with long-term climatic/oceanographic changes (Kim et al., 2001). Over the last two decades (1982 – 2006), a rapid increase of sea surface temperature (SST) of

1.09 ℃ has been recorded in the East Sea, which is the fourth highest among the 18 large marine ecosystems in the world ocean (Belkin, 2009). Increased SST reduces the soubility of

$O_2$ in the surface mixed layer and enhances stratification, which ultimately affects biological production in the water column and suppresses transport of $O_2$-rich surface water into the deep bottom. Indeed, recent oeanographic 
[revised manuscript text omitted]

---

## Author Comment (AC3) · 7 Oct 2016

Response on the interactive comment on "Manganese and iron reduction dominate organic carbon oxidation in deep continental margin sediments of the Ulleung Basin, East Sea" by Jung-Ho Hyun et al.

Anonymous Referee #3 The zonation of terminal electron accepting processes (TEAPs) in marine sediments and the importance of anaerobic TEAPs to the overall degradation of deposited organic carbon are concepts that all students of biogeochemistry learn very early in their careers. Nevertheless, as the authors point out, there are very few studies that attempt to quantitatively estimate the contributions of the various anaerobic dissimilatory processes to overall organic

carbon decomposition. Employing a carefully calibrated set of biogeochemical rate measurements Hyun et al. tease apart the contribution of Mn, Fe and sulfate reduction to organic carbon degradation in the Ulleung Basin of the East Sea bordered by Korea, Russia and Japan. This builds on a research approach pioneered in the nineties by Thamdrup, Canfield and co-workers. Here, Hyun et al. show that Mn reduction can be a powerful and important TEAP in during sedimentary organic matter degradation. While there are a number of assumptions built into the experiments used to estimate dissimilatoryFe and Mn reduction rates, this is still state of the art approach in sedimentary biogeochemistry. Moreover, from such experiments, they conclude that Mn reduction may be underestimated as a process in marine sediments as the techniques for studying Fe and Mn reduction require thorough and careful experimentation, as demonstrated in this study. Hyun et al.'s study is a further, solid step in the right direction and stands in stark contrast to recent efforts that only examine pore water concentration profiles (e.g. Bowles et al., Science 2014), which underestimate or miss the contribution of Fe, Mn and sulfate reduction towards organic carbon decomposition in near-surface sediments. I especially appreciated the attempt to link the relative contribution to total carbon oxidation as a function of the Mn content of the sediment. In general, this is a solid contribution to marine sediment biogeochemistry and I look forward to using these results as an excellent example of anaerobic TEAP processes in classes for the next generation of biogeochemistry students. (General response): 1. Thank you very much for this highly positive comments. We will incorporate your comment in revised manuscript as much as we can. 2. In this revised manuscript, we have modified the title of the manuscript to: "Manganese and iron reduction dominate organic carbon oxidation in surface sediments of the deep Ulleung Basin, East Sea" to clarify that the dominance of Mn and Fe reduction occurs in the surface sediments of the deep basin.

Specific comments:

1) Line 91: The authors might want to consider estimates from D'Hondt et al.

Science 2004, where areal rates of Mn and Fe reduction in the Peru Basin have been estimated. Although derived from deep sub-surface pore water profiles, it would be interesting to see how the Peru Basin site maps onto Figure 7. (Response): Thank you for this suggestion. I agree that it will be nice if it fits onto Figure 7. The contribution of Mn reduction in total anaerobic Corg oxidation in the Peru Basin was 60% (see Table 1 in D'Hondt et al. 2004). The MnO content in the same site was 0.21 wt.% (see, http://www-odp.tamu.edu/publications/201_IR/chap_12/c12_t2.htm#1005989, Initial report of ODP Leg 201), that is equivalent to 46 ïĄ∎mol cm-3. So, if we incorporate the value of the Peru Basin (PB) into the Fig. 7, it comes out as below;

It looks good at first glance. Please note, however, that the other data in the Fig. 7 is derived from the contribution of Mn reduction in a given depth interval in the sediment compared to the DCA-extractable Mn content in the same interval, with both parameters changing with depth in the sediment. The Panama Basin value, however, represents a depth-integrated value. Thus, the numbers are not really directly comparable, and we may have to decide not to add the point to the graph in the manuscript. Thank you again for adding an idea, and it was exciting to execute your idea. You may use the plot that we produced above. 2) Line 169: Pore water analysis....can approximate detection limits be provided? (Response): We have provided the detection limit or reproducibility of each constituent as you recommended in line 185 - 187 in the revised manuscript as follows "The detection limit of H2S, Ca2+, Mn2+ and Fe2+ was 3 ïĄ∎M, 1.8 ïĄ∎M, 3 ïĄ∎M and 1 ïĄ∎M, respectively. Reproducibility of DIC and NH4+ was better than 10%. Precision of NO3- was 1 – 2%."

3) Line 249 and following : It would be helpful to have a formula that includes the effect of SR the estimate of dissimilatory Fe (III) reduction (Equation 4) that includes the stoichiometry from Equation 5. (Response): The procedure used was adopted from Gribsholt et al. (2003). The rationale is that in the presence of reactive Fe(III), H2S produced from sulfate reduction reacts with Fe(III) and the Fe2+ produced from this reaction precipitates as FeS. The stoichiometry of abiotic reduction of 2Fe coupled to reaction of 3H2S in marine sediments is presented in several studies. For example, Melton et al. (2014, Nature Reviews. Microbiol.12: 797 – 808) stated that "At neutral pH, hydrogen sulphide (H2S) can abiotically reduce Fe(III) oxyhydroxides: 2FeOOH + 3H2S → 2FeS + S0 + 4H2O. H2S reactions with Fe are especially important in marine environments, where high sulphate concentrations and microbial S reduction lead to pronounced H2S production". So, the H2S in the equation 5 represents the H2S produced from the sulfate reduction that we directly measured using 35S injection incubation exeriment.

(Correction): We have changed the equation (5) to "2FeOOH + 3H2S(produced by SR) = 2FeS + So + 4H2O, and add one more review paper as a reference (Melton et al., 2014) in line 256 – 259 in revised manuscript.

4) Line 376...."that" does not take a comma. (Response): I removed the comma (line 385).

5) Line 388 and 414: Are "evidenced" and "zonated" proper verbs? (Response): We will change the 'evidenced that ∼∼" to "presented evidence that ∼∼" (line 397). We will aso change the "systematically zonated with discrete sequential depletion according to the order of decreasing energy yield for Corg oxidation" to "systematically distributed with discrete zonation (line 422) according to the order of decreasing energy yield for Corg oxidation"

6) Line 440 "Consequently" not necessary (Response): I will remove the "Conequently" in revised manuscript (line 448).

7) Line 489 Replace "this" with "these" (Response): Yes. I will replace it in revised manuscript (line 510).

8) Paragraph at line 498: I find this argument to be a bit of a stretch. Basically the authors are saying because Canfield et al. had a negative result with a complexed Fe experiment that they also have no effect of ferrous iron on MnO4 reduction. This is a bit weak. (Response): Thank you for this comment. In addition to the experimental results by Canfield et al., we also discussed that the MnO2 reduciton is a favorable Corg oxidation pathway that produces Mn2+ in line 526-527).

9) Paragraph starting at Line 548: I found this part very interesting. What happens when one moves out into total oxic sedimentary environments where oxygen penetration is deep and there are plenty of Mn oxide crusts (deep ocean sediment)? Should perhaps the Figure 7 refer to relative contribution of Mn reduction to "anaerobic" carbon oxidation? (Response): Thank you for this comment. Yes! It is a contribution of Mn reduction to anaerobic C oxidation. To avoid the confusion, I changed the figure caption of the Figure 7 as follows "The relative contribution of Mn reduction to anaerobic carbon oxidation as a function of the concentration of Mn(DCA) at 3 different sites. BS, Black Sea (Thamdrup et al. 2000); UB, Ulleung Basin (This study); Sk, Skagerrak (Canfield et al. 1993b)" We also replaced "total" with "anaerobic" in line 583.

10) Line 594: The authors might point out that they are probably underestimating total oxygen uptake. Bioirrigation might very likely play a major role in these sediments. (Response): It is correct that bioirrigation can contribute substantially to total oxygen uptake in some sediments. Bioirrigation has not been investigated in the UB sediments. We have discussed the potential underestimation of oxygen uptake and the consequences for the conclusions based on the large dataset reviewed by Glud (2008; Mar Biol Res 4: 243-289).

1. Finally, we have corrected our quantative estimation on the partitioning of Corg oxidation pathways at each station in line 476 – 488: "Additionally, our partitioning of carbon oxidation pathways could be biased towards the anaerobic electron acceptors due to the use of the diffusive oxygen uptake (DOU) rather than total oxygen uptake (TOU), which will exceed DOU if bioirrigation is active (Glud 2008). Bioirrigation was not determined at our sites, but the pore water profiles show no indication of strong irrigation (Fig. 2). An average DOU/TOU ratio of ∼0.6 has been reported for sediments at 1.5 – 2.5 km depth (Glud 2008). Using this ratio, and assuming that TOU is partitioned similarly as DOU between aerobic carbon oxidation and reoxidation, aerobic carbon oxidation would account for 25%, while Fe and sulfate reduction would account for 11% and 46% of of carbon oxidation, respectively. This, the potential bias from using DOU is not expected to affect the ranking of electron acceptors by quantitative importance (SO42- > O2 > Fe(III)), and, as discussed further below, the partitioning of Corg oxidation at M1 falls within the range previously reported for continental margin sediments."

2. We also discussed the significance of aerobic respiration at basin site (D3) in line 505 - 507 in revised manuscript: "Correction for a potential underestimation of TOU, as discussed for M1, would reduce the contributions of Mn and Fe reduction slightly to 41% and 18%, respectively."

Please also note the supplement to this comment:
http://www.biogeosciences-discuss.net/bg-2016-222/bg-2016-222-AC3-supplement.pdf

[Figure]

**Fig. 1.**

**Supplement:**

[revised manuscript text omitted]

thermohaline convection system that is driven by the high density surface water sinking (Kim
et al., 2001) in a manner similar to that of the Great Ocean Conveyor Belt (Broecker, 1991).
The turnover time (ca. 100 – 300 years) of the thermohaline circulation is shorter than that of
the global conveyor belt of 1000 – 2000 years (Broecker and Peng, 1982). Because of the
shorter time-scale, together with the relatively small volume, the East Sea is expected to be
much more sensitive to global environmental changes (such as global warming) compared
with the open oceans. In this regard, the East Sea has been considered as a natural laboratory
that provides a useful field for large-scale oceanographic experiments to predict the response
of oceans associated with long-term climatic/oceanographic changes (Kim et al., 2001). Over
the last two decades (1982 – 2006), a rapid increase of sea surface temperature (SST) of
1.09 ℃ has been recorded in the East Sea, which is the fourth highest among the 18 large
marine ecosystems in the world ocean (Belkin, 2009). Increased SST reduces the soubility of
$O_2$ in the surface mixed layer and enhances stratification, which ultimately affects biological
production in the water column and suppresses transport of $O_2$-rich surface water into the

[revised manuscript text omitted]

---

## Author Response (AR1)

**Associate Editor Decision: Publish subject to minor revisions (Editor review)**

(23 Oct 2016) by Dr. Silvio Pantoja
Comments to the Author:
October 23, 2016

Dear Dr. Hyun,

Thanks for providing responses to three Reviewers of your BG discussion paper (bg-2016-222). I would like to invite you to submit a revised version of the article based on your responses, and considering the following issues:

1) Reviewer 1: Question 4) Line 606: "The statement about the probable importance of bioturbation seems to be in contradiction with the well-defined utilization of the electron acceptors according to the order of decreasing energy yield for organic C oxidation that has been underscored in lines 412-417? Again, I suggest clarifying this point."

Reviewer 1 asked clarification to the following: There is a clear biogeochemical zonation in these sediments (lines 412-417) and your response agreed with that, but still in line 606 it says "Thus, it is realistic that bioturbation drives Mn cycling in the UB. ". To me is contradictory with lines 412-417 as well, unless you meant something else. Please clarify that and proceed accordingly in the revised version.

**(Response):** To clarify the systematic zonation of the electron acceptor at D3 where bioturbation derives Mn cycling, we have added a paragraph in line 628 – 635 as follows: "Meantime, the estimated biodiffusion coefficient of (Db) of 9.5 cm$^2$ yr$^{-1}$ at Site D3 corresponds to ~2% of the molecular diffusion coefficient of oxygen (388 cm$^2$ yr$^{-1}$ ). Judging from the absence of major fauna in the UB sediments, the mixing is brought about by small organisms with each individual affecting only a small area relative to the size of our cores, and the Db averaging many of these small but frequent events. Therefore, we see no contradiction between the presence of bioturbation and the relatively distinct redox zonation at D3 (Fig. 5F). Similarly, Hyacinthe et al. (2001) found that well defined profiles can be observed in both sediments with low and high bioactivity in the Bay of Biscay."

2) Reviewer 1: Minor 4) Line 276: I suspect that the units (ml/g) are erroneous?

(Response): "It is presented in Thamdrup et al (2000). "

Something being published cannot be a proper response to a colleague reviewer. What are unit ml/g of?

**(Response):** I am sorry for the inappropriate response. I was even wrong in citing the reference by stating Thamdrup et al. (2000). It was explained in Canfied et al. (1993b) and Thamdrup and Dalsgaard (2000). Here is our response. If you see the following figure (the Fig. 7 in Canfield et al. 1993b, GCA), the unit is derived from the slope of Mn adsorption experiments (= μmol g$^{-1}$ / μM = 10$^3$ ml g$^{-1}$) in the Skagerrak. Those references (Canfield et al., 1993b; Thamdrup and Dalsgaard, 2000) are listed in the line 282 in the revised manuscript.

[Figure]

FIG. 7. Figure shows results from Mn adsorption experiments for sediment from various depths at S$_9$, and 1–2 cm at S$_6$. Included also are results from desorption experiments (see text).

3) Reviewer 2." 40) L. 638: I still do not understand why the Ulleung Basin is a

"biogeochemical hotspot"? Is it because organic matter mineralization is dominated by metal reduction? This is not clear at all and I would therefore suggest to better explain or to delete this.

**(Response):** It is not because organic carbon mineralization is dominated by metal reduction. The reason that we stated the UB as a biogeochemical hot spot is that the overall organic carbon oxidation in the UB is higher than those measured in major up- welling system such as Benguela upwelling system and is even comparable to those reported at the continental slope of the Chilean upwelling system at a similar depth range of 1000 – 2500 m. Please see the line 639 – 661 in the revised manuscript. "

Please demonstrate that Ulleung Basin is a "biogeochemical hotspot" showing numbers
to compare in the revised version.

**(Response):** To demonstrate the UB as a biogeochemical hotspot, I have added the
number of SRRs reported in the Benguela upwelling system ($0.14 - 1.39$ mmol m$^{-2}$ d$^{-1}$),
Chilean ($2.7 - 4.8$ mmol m$^{-2}$ d$^{-1}$) and Peruvian upwelling system ($5.2$ mmol m$^{-2}$ d$^{-1}$) in
line $640 - 643$ in revised manuscript..

I agree with Reviewer 2 (S Kasten) that there is no need of spending time/space
highlighting this issue since uniqueness of your scientific contribution is what matters
here.

**(Response 1):** As you and the reviewer #2 pointed out, I agree that there is no need of
spending too much time/space highlighting this issue. So, we have substantially curtailed
the length of the paragraph by deleting the following 16 lines "The East Sea is often called
as "a miniature ocean" because of the independent thermohaline convection system that is
driven by the high density surface water sinking (Kim et al., 2001) in a manner similar to that
of the Great Ocean Conveyor Belt (Broecker, 1991). The turnover time (ca. $100 - 300$ years)
of the thermohaline circulation is shorter than that of the global conveyor belt of $1000 - 2000$
years (Broecker and Peng, 1982). Because of the shorter time-scale, together with the
relatively small volume, the East Sea is expected to be much more sensitive to global
environmental changes (such as global warming) compared with the open oceans. In this
regard, the East Sea has been considered as a natural laboratory that provides a useful field
for large-scale oceanographic experiments to predict the response of oceans associated with
long-term climatic/oceanographic changes (Kim et al., 2001). Over the last two decades
($1982 - 2006$), a rapid increase of sea surface temperature (SST) of $1.09$ °C has been
recorded in the East Sea, which is the fourth highest among the 18 large marine ecosystems
in the world ocean (Belkin, 2009). Increased SST reduces the soubility of $O_2$ in the surface
mixed layer and enhances stratification, which ultimately affects biological production in the
water column and suppresses transport of $O_2$-rich surface water into the deep bottom."

**(Response 2):** Nonetheless, we still think it is important to mention the UB as a
biogeochemical hotspot in this manuscript. In two previous papers (Lee et al. 2008;
Hyun et al. 2010), we have argued that the sediment of the UB is a place where benthic
mineralization is exceptionally high, considering the water depth, due to the formation of
highly productive upwelling conditions in overlying water column. Based on the repeatedly high benthic mineralization rates in present study together with the previous
results, we feel that this distinct aspect of the UB deserves mentioning in line 639 – 661
in revised manuscript, and we find that the term "biogeochemical hotspot" captures this
well. We also believe it is important to stress shortly the significance of monitoring the
variations of $C_{org}$ oxidation pathways since the DO in the bottom water of the UB has
been decreasing ~10% over the last 30 years as stated in line 662 – 671.
I sincerely hope this revision is acceptable for you.
4) Reviewer 3: 8). I suggest moving evidence in lines 526-527 to paragraph starting in
line 498 to support your argument.
(Response): Thank you for the suggestion. I moved the sentence "As manganese
reduction is thermodynamically more favorable than iron and sulfate reduction, the $Mn^{2+}$
liberation (Fig. 4) is likely resulted from dissimilatory Mn reduction." to line 510 – 511 as
you suggested.
Looking forward to hearing from you
Sincerely yours
Silvio Pantoja
Associate Editor
Thank you again for your time.
Jung-Ho Hyun

---

## Author Response (AR2)

**Associate Editor Decision: Publish subject to minor revisions (Editor review)** (07 Jan 2017) by Dr. Silvio Pantoja

Comments to the Author:
7 January 2017
Review 2

Dear Dr. Hyun,

Thanks for your responses.

The comment below:
1) Reviewer 1: Question 4) Line 606: "The statement about the probable importance of bioturbation seems to be in contradiction with the well-defined utilization of the electron acceptors according to the order of decreasing energy yield for organic C oxidation that has been underscored in lines 412-417? Again, I suggest clarifying this point."

Reviewer 1 asked clarification to the following: There is a clear biogeochemical zonation in these sediments (lines 412-417) and your response agreed with that, but still in line 606 it says "Thus, it is realistic that bioturbation drives Mn cycling in the UB. ". To me is contradictory with lines 412-417 as well, unless you meant something else. Please clarify that and proceed accordingly in the revised version.

Refers to this:
If there is zonation, then bioturbation is not high enough otherwise biochemical zonation would disappear (or masked). This is the contradiction raised by Reviewer 1, which is not clarified in your response since the statement "Thus, it is realistic that ⌜SEP⌝ bioturbation drives Mn cycling in the UB" is still there.

Please revise accordingly

Sincerely yours
Silvio Pantoja
Associate Editor

**Response to the 2nd comment by associated editor on bg-2016-222 (07 Jan 2017)**

Dear Dr. Pantoja:

Thank you again for your comments on our revision. We had addressed the concerns about bioturbation vs. distinct chemical zonation in a new paragraph right after the statement about bioturbation as a realistic mechanism. However, from your comment and from when reading the text again, I realize that format of previous version could be confusing. According to your suggestion, we have revised the line 624 – 634 in previous version as follows: "This value is 3.6 times lower than the coefficient estimated for the Skagerrak (Canfield et al., 1993b) and consistent with estimates for other sediments with similar deposition rates (Boudreau, 1994). The estimated biodiffusion coefficient (Db) of 9.5 $cm^2$ $yr^{-1}$ at Site D3 corresponds to ~2 % of the molecular diffusion coefficient of oxygen (388 $cm^2$ $yr^{-1}$). Judging from the absence of major fauna in the UB sediments, the mixing is brought about by small organisms with each individual affecting only a small area relative to the size of our cores, and the Db averaging many of these small and local but frequent events. Under such conditions, bioturbation can drive Mn cycling in the UB without substantial smearing of the redox zonation. Similarly, Hyacinthe et al. (2001) found that well defined profiles can be observed in both sediments with low and high bioactivity in the Bay of Biscay."

Finally, I sincerely hope this revised version is acceptable for you. Thank you again for your precious time to improve the quality of this manuscript.

Best regards,

Jung-Ho Hyun